# Historical droughts in the Qing dynasty (1644-1911) of China

Kuan-Hui Elaine Lin[1], Pao K. Wang[2,3], Pi-Ling Pai[4], Yu-Shiuan Lin[2] and Chih-Wei Wang[2]

[1]Graduate Institute of Environmental Education, National Taiwan Normal University, Taipei, Taiwan
[2]Research Center for Environmental Changes, Academia Sinica, Taipei, Taiwan
[3]Departent of Atmospheric and Oceanic Sciences, University of Wisconsin-Madison, Madison, Wisconsin, USA
[4]Research Center for Humanities and Social Sciences, Academia Sinica, Taipei, Taiwan

*Correspondence to*: K-H E. Lin (khelin@ntnu.edu.tw); Pao K. Wang (pkwang@gate.sinica.edu.tw)

**Abstract.** This study presents a new epistemological analysis of drought chronology through a well-defined methodology for reconstructing past drought series as well as series of other associated ecological and societal impact variables. Instead of building grading system based on mixed criteria, this method facilitates transparency in reconstruction process and enables statistical examination of all variables when building series. The data for the present study is derived mainly from the REACHES database, however, other archival documentary and index data from independent sources are also applied to understand drought narratives and to cross check and validate the analysis derived from the REACHES. From time series analysis, six severe drought periods are identified in the Qing dynasty, and then spatial analysis is performed to demonstrate spatial distribution of drought and other variables in the six periods as well as social network analysis to reveal connections between drought and other ecological and societal variables. Research results clearly illustrate the role of human intervention to influence the impacts of drought on societal consequences. Particularly, the correlation between drought and socioeconomic is not strong; crop failure and famine are important intermediate factors, meanwhile ecological factor such as locust and disaster relief measures are all imperative to intervene between crop production and famine. Implications of the study on drought impact are provided as well as its significance on historical climate reconstruction studies.

## 1. Introduction

Global warming is expected to influence the earth's hydrological cycle and put higher stress on water resources. Climate simulations reveal a wide expansion of dry areas over land and an intensification of wet-dry contrast at lower latitudes (Held and Soden, 2006; Schewe et al., 2014). There are some recent observations that hint agreement with the projected enlarging wet-dry contrast pattern and intensified drought (Schewe et al., 2014) although substantial uncertainties about specific patterns and trends remain. Even less understood is the interannual and decadal scale change pattern of regional precipitation and the duration and magnitude of drought at finer spatial and temporal resolution (IPCC 2013). Obviously, more high quality hydroclimate data are necessary if we wish to remove these uncertainties. Such data are also necessary for assessing the performance of climate models, especially on their capability in projecting reliably the climates of next few decades. The length of instrumental meteorological data series is only a little more than a hundred years which is not long enough to assess the validity of climate models that make predictions of many decades into the future (Cook et al., 2014; Ljungqvist et al., 2016).

The most direct manifestation of hydroclimate variability is the drought and flood of the environment (Routson et al., 2016; Stevenson et al, 2018). Drought and flood are two extremes of the hydrological cycle and both can induce severe environmental and socioeconomic consequences. Flood which generally represents sudden onset of excessive water can lead to immediate loss of lives and material damages. The formation of drought, on the contrary, normally refers to a slower and longer water deficiency process that can result in a more chronologically extended range of impacts on agriculture and water scarcity contributing to catastrophic socioeconomic outcomes (Brázdil et al., 2019). Given the intensified drought condition in the future scenarios as described above, there is an urgent need to study past drought, especially the severe drought cases, to decipher their occurrence, duration, magnitude, and the associated ecological and societal consequences for implications of future adaptation.

Studying past drought and humidity has been a long practiced subject in historical climatology and paleoclimatology (Stahle et al., 2007; Tan and Liao, 2012; Yi et al., 2012; Cook et al., 2015; Ge et al., 2016; Hao et al., 2016; Brázdil et al., 2018; Shi et al., 2018). In the instrumental era, drought is relatively easy to be defined based on the temperature and precipitation measurements. The instrumental era can be tracked back to the mid or late 19$^{th}$ century when many national weather services started (Broennimann et al., 2018). Some early instrumental meteorological data might be acquired in some regions dated back to the late 17$^{th}$ and early 18$^{th}$ century. Prior to the instrumental era, there are primarily two sources of data that can be used to build drought series of high temporal resolution: tree ring data and documentary records (White, 2019). Tree rings provide abundant information about moisture, and sometimes temperature, for reconstructing millennial long hydroclimate indices from which drought can be inferred. The most well-known works include Old World Drought Atlas using tree ring data (Cook et al., 2015) calibrated with Palmer Drought Severity Index (PDSI) (Palmer, 1965), Monsoon Asia Drought Atlas (Cook et al., 2010), and Megadroughts over North America (Stahle et al., 2007). Yet, tree ring reconstruction often suffers from the uncertainty in growing seasonality of trees and ambiguous interpretation of isotopes (Schofield et al., 2016). Nonetheless, high quality quantitative drought-indices series such as Standardized Precipitation Index (SPI) (Mckee et al., 1993), Standardized Precipitation-Evapotranspiration Index (SPEI) (Vicente-Serrano et al., 2010), PDSI, and Palmer Z Index can also be reconstructed from documentary data. For example, Brázdil et al., (2016) and Možný et al. (2016) have built monthly, seasonal, half-year and annual resolution SPI, SPEI, Z-index, and self-calibrated PDSI for the Czech Lands. Moreover, some other special drought indices series were developed in Europe like Drought Rogation Index in Spain (Barriendos, 1997) or Drought Index in Italy (Diodato and Belocchi, 2011).

Documentary records reflect direct human observations of weather condition and are less ambiguous when used for reconstructing past climate. They contain weather, climate, phonological, and socioeconomic situations in specific locations in various time resolution such as daily, monthly or seasonal, allowing comprehensive analysis on drought events, scales, and contexts (Brázdil et al., 2019; Huang et al., 2019). To be sure, there are also some criticisms about inferring drought from documented records, for example, it is sometimes hard to distinguish whether the drought record refers to rainfall deficiency, hydrological process or crop irrigation need. Several studies tried to remedy this by proposing four categories of drought: meteorological drought caused by significant reduction of precipitation for weeks or months compared to normal precipitation; agricultural drought associated with lack of water for plant growth

for a period lasting from weeks to 6-9 months; hydrological drought characterized by a shortage or absence of water in water courses, reservoirs or aquifers; and socioeconomic drought caused by negative effect of drought on everyday life, social system and stability (Heim, 2002; Brázdil et al., 2018). There is yet another view suggesting drought as a complicated process of water shortage caused and modified by human processes (Van Loon et al., 2016). Despite of such shortages, documentary records are still one of the most important sources for reconstructing drought series, only that we have to be very careful in extracting proper information from them.

Brazdil et al. (2018) have described various types of documentary records that can be used for extracting drought information. Before the instrumental era, all historical climate records are essentially qualitative or narrative in nature, and it is necessary to transform them into numerical data series so that statistical analysis of them can be performed. The most common practice is to design a three-, five- or seven-grade system based on the duration and severity of drought (Zheng et al., 2014; Brazdil et al., 2018) so that every single quotation of drought record can be assigned a specific value according to the grading criteria. Arranging these grades as a sequence in time and we get a time series of drought grade, often called a drought index series. In reality, however, transformation process from qualitative narratives to digital series can be tricky because a wide spectrum of grade assignment strategy exists and different investigators can derive very different index series based on the same set of drought records. More detailed discussions on this matter have been given by Brázdil et al. (2005) and Pfister et al. (1999).

In China, a dryness-wetness (also referred to as drought/flood) index has been widely used in recent decades for reconstructing past climates (Tan and Liao, 2012; Yi et al., 2012; Zheng et al., 2014, Hao et al., 2016; Shi et al., 2018, for detailed review on historical climate records in China and reconstruction of past climates please refer to Zhang and Crowley 1989). The index series was initially built by the Academy of Chinese Meteorological Science (CMA, 1981; Wang and Zhao, 1981) based on historical documentary records and was sequentially expanded by several Chinese scholars (see Zhang, 2003; also see Hao et al., 2019). The index system, based on Yi et al. (2012) with modifications from Zheng et al. (2014) and Tan et al. (2013), specifies a five-grade scheme as described in Table 1:

**Table 1: Dryness-Wetness index in China**

| Grade | Severity | Examples of some descriptions |
|-------|----------|-------------------------------|
| 1 | Very wet | Prolonged heavy rain (e.g. excessive rain continued over a month in spring or summer), Extensive flood (e.g. heavy rain for several days, land flooded, boating on land), Unusually heavy typhoon rain (e.g. cropland and houses of several counties inundated by typhoon rainfall) |
| 2 | Wet | Spring or autumn prolonged rain with moderate damage (e.g. protracted rain in spring or autumn), local flood (e.g. counties flooded for months, drought in spring but heavy rain in summer) |
| 3 | Normal | Favorable weather (e.g. good weather for crops, bumper harvest) |
| 4 | Dry | Light drought disaster in single season (e.g. drought in spring or autumn), local light drought disaster (e.g. short rainfall in summer, drought in other months, |

| 5 | Very dry | Continued drought for more than one season or for several months (e.g. drought from spring to summer or from summer to autumn, no rain in four months and rivers dried out), severe drought over extensive area (e.g. thousands of miles of barren land, severe drought throughout south of the Yangtze River lower reach), famine (e.g. tens of thousands starved to death on the street, hungry people consumed tree roots or fine soils) |
| --- | --- | --- |

This scheme uses mixed meteorological/hydrological/socioeconomic criteria to determine the drought grade to form China's dryness-wetness index series. This system has the advantage in evaluating drought severity, however, it is difficult to identify specific drought categories in the records as mentioned earlier. Thus one cannot identify whether or not a certain drought event was associated with meteorological deficiency of rainfall or had more to do with the failure of the agricultural irrigation systems. This can lead to biased interpretation of drought records as representing the atmospheric humidity condition. It also prevents proper statistical associations among different drought categories (each as an independent variable) and could lead to erroneous conclusions about the nature of the drought and its impact.

This study takes a new conceptual point in examining the drought records and proposes a new method to reconstruct past drought series and that of other associated ecological and societal variables from Chinese historical documentary records. Our objective is to make the interpretation of every drought and associated variables as literally clear and operationally independent as possible. This method can facilitate statistical examinations of these variables and enable transparency in the reconstruction process of historical drought series. Moreover, we will also perform contextual analysis of severe drought events through analyzing the strength of relation among different variables and at the same time examining their spatial characteristics. In order to assess more clearly the true nature of the drought event, we acquired more archival information from historical documents so as to gauge governmental and societal responses to severe drought events.

In the following sections, we will first briefly introduce the data used for analysis (Sect. 2). Then the methods will be described (Sect. 3), followed by the presentation of the results of time series analysis of those variables. We will then examine the six severe drought periods, their spatiotemporal patterns and the details of the narratives (Sect. 4). Finally, the climate-society interaction will be explored, followed by an outlook into the future research work (Sect. 5).

## 2. Data
### 2.1 Drought and associated data from REACHES database

The main data source of this study is the REACHES database which is a digitized and category-coded climate records
in Chinese historical documents in the last three millennia (readers are referred to Wang et al., 2018 for detailed
descriptions about the database). The present study utilizes only the records in the last imperial dynasty of China, i.e.,
the Qing dynasty (1644-1911). There is a total of 93,415 records (database version VOL34-V3.1-04-E2) in this period
and we retrieved drought and related records including locust, dried waterbody (representing hydrological drought),
crop failure (representing agricultural drought), famine and socioeconomic turmoil (along with famine to represent
socioeconomic drought). Locust outbreak is an important ecological indicator closely related to drought and famine
as locust outbreak often occurs in drought climate and these insects consume great quantities of crops resulting in
wide spread famine and even cannibalism (Huang et al., 2019). Rigorous statistical study of locust outbreaks and their
relation with drought and societal aftermath they caused are rarely performed.
REACHES has a sophisticated code system to include as much information as conveyed in the historical records. It
uses a 9-digit code scheme: the first two digits describe the main category of the climate nature (for example,
precipitation, temperature, etc.). The next two digits describe the subcategory (for example, rainfall and snow are two
of many subcategories under precipitation category) and the next three digits describe special vocabularies (for
example, light rain and torrential downpour under rainfall subcategory). The last two digits denote the magnitude and
time information. As an example, a code of a record 300130156 can be deciphered as follows: the first two digits "30"
indicate that the record belongs to the main category of drought, the next two digits "01" indicates that it is in the
"drought" subcategory (i.e., the record says "drought" directly). There are other subcategories related to drought but
were described in different contexts, for example, "watercourse dried out" is assigned as subcategory 11. The next
three digits "301" indicate that its vocabulary is major drought (亢旱, *Kang Han*). The last two digits "56" denote
magnitude and time-duration information. In this case, 5 means the magnitude is "heavy and occurred many times"
and "6" indicates the event duration lasted between 60 and 90 days.
Famine is also a main category item with the code 35. On the other hand, locust outbreak is a subcategory (code 01)
item under the main category of pest/vermin category (code32). Thus the complete code representing a locust record
is 3201. Crop is a main category item (code 33) that comprises various degrees of harvest conditions and crop species,
and a vocabulary code system ranging from 100 to 499 is assigned to describe them individually. Similarly,
socioeconomic turmoil is a main category item with code 71 and the subcategory items most relevant to this study
include immigration/displacement (code 03, so the total retrieval code is 7103), battle/war (code 05), impoverishment
(09), death/severely injured (code 10), human trafficking (code 12), and abandoned settlements (code 16).
As the historical documents were written by a great variety of author types, not all of them were written in a fixed
standard style and special care must be taken to denote and encode these records. For example, some records
sometimes use a negative tense to describe the absence of a phenomenon which should take place around the time,
such as "no drought" (無旱, *Wu Hang*). In this case, it would be encoded under drought category with magnitude code
8 denoting "not occurring". Since "no drought" items should not be taken into the drought series in this study, we
have carefully removed all those records with magnitude code 8 in the data series.

Finally, and importantly, all records of the REACHES are taken from the *Compendium of Chinese Meteorological Records of the Last 3,000 Years* (Zhang, 2013), which includes only records that have direct (or indirect) implications or linkages with meteorological phenomena. Those not directly related to meteorological matter were not included in the *Compendium*. In particular, some socioeconomic events that are not explicitly linked to climate conditions may not be included in the *Compendium* but, as often happens in human society, they may be linked in some indirect way. This means that care must be taken when using and analyzing the socioeconomic variables in the REACHES because of the potentially biased sampling. Other sources of relevant socioeconomic data should be considered to cross check and validate the analysis.

We retrieved 51,974 records related to drought and other related variables from the REACHES database (version VOL34-V3.1-04-E2) for the period of 1644-1911. Figure 1a shows time evolution of records for the variables of drought, locust, crop failure, famine and socioeconomic turmoil, and Fig. 1b shows number of sites (i.e. counties, cities and prefectures, in REACHES. Total n of sites with drought records=1,404) that have the corresponding records of the year. The two diagrams exhibit almost identical trends showing good general consistency of the data series of these variables and it is also highly likely that the records are evenly or randomly distributed over the sites. All data derived from the REACHES database pertinent to this study will be deposited at NOAA national center for environmental information (https://www.ncdc.noaa.gov/paleo-search/study/23410).

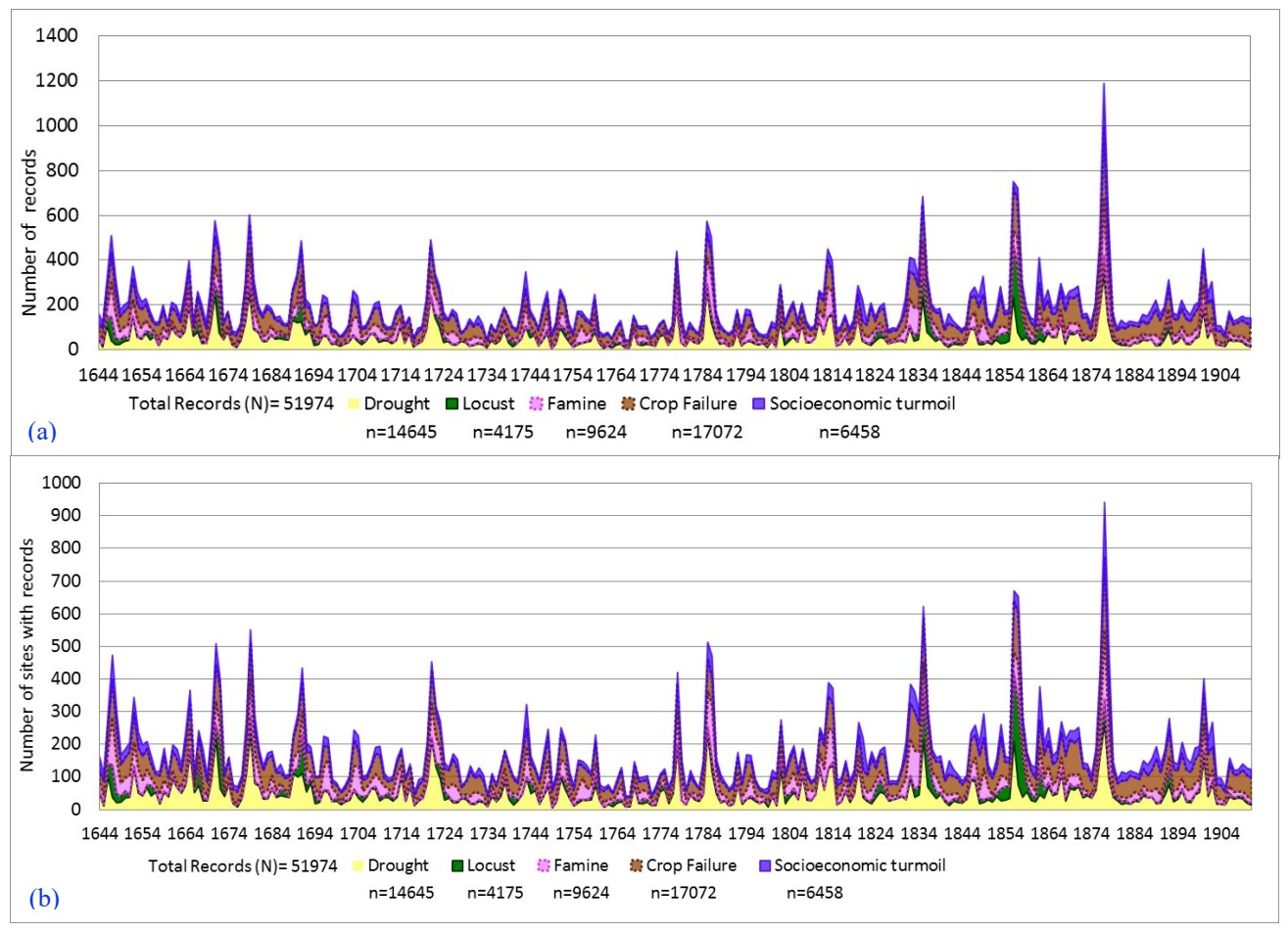


**Figure 1: Time series plots of drought and associated series derived from the REACHES database. (a) shows**
**number of records per year; (b) shows number of sites (counties, cities or prefectures) that have records of the**
**year. Thus, one site may have more than one record for the year. Note that drought variable here includes**
**meteorological and dried waterbody related hydrological drought, crop failure is interpreted as agricultural**
**drought, famine and socioeconomic turmoil are taken to represent socioeconomic drought.**

**2.2 Other archival and index data**

As mentioned previously, the *Compendium* does not focus on and hence collect in a comprehensive manner
socioeconomic events, therefore sampling bias may exist. To comprehensively compare and analyze drought and
associated data series from the REACHES with socioeconomic variables, we have consulted with independent
documentary sources aside from the *Compendium*. These include Draft History of the Qing Dynasty (清史稿, *Qingshi*
*Gao*), Actual Veritable Records of Emperors of the Qing Dynasty (清實錄, *Qing Shilu*), Gazetteers of Xunzhou
Prefecture (Tongzhi reign edition) ( [同治]潯州府志, *[Tong Zhi] Xun Zhou Fu Zhi*), Gazetteers of Gui County
(Guangxu reign edition) ([光緒]貴縣志, *[Guang Xu] Gui Xian Zhi*), Gazetteers of Pingnan County   (Guangxu reign
edition) ([光緒]平南縣志, *[Guang Xu] Ping Nan Xian Zhi*), and   Gazetteers of Yongchun County (Chinese
Republican Period edition) ([民國]永淳縣志, [Min Guo] *Yong Chun Xian Zhi*). Practically, it might be unavoidable
that some of the contents in the historical books, if referred to climatic and weather conditions, were quoted in the

*Compendium*. For example, there are 5 quotations of records from Actual Veritable Records of Emperors of the Qing Dynasty and 148 records from Draft History of the Qing Dynasty found in the REACHES among the overall of 93,415 records in the Qing dynasty. The coverage, however, is minimal in quantity among all, in the meantime, these documents contain meticulous records of social, economic and institutional events of the time. Local gazetteers particularly contain detailed information about social unrests such as the massive rebellion, Taiping Civil War (太平天國 [Tai Ping Tian Guo]), in the mid-19th century.

Besides the above documents, we also collected records on grain price, civil war and population indices for further analysis. Grain price data (1738-1911) was derived from Qing Dynasty Grain Price Database which compiles the monthly grain price reports preserved in the First Historical Archives of China in Beijing and the National Palace Museum in Taipei by Wang (2009). Civil war data that reveals the frequency of civil wars in Qing Dynasty was derived from Chronology of China's Ancient Wars by Chinese Military History Writing Group (1985). Population index data comes from several different sources: Provincial population statistics 1661-1776 CE from Encyclopedia of Official Documents of the Qing Dynasty (清朝文獻通考, *Qingchao Wenxian Tongkao*); Provincial population statistics 1780-1890 from Annual Registers of Quantities of Provincial Population and Grain Storage by the Ministry of Revenue (戶部匯奏各省民數穀數清冊, Gesheng Minshu Gushu Qingce); and The History of Population in China, Vol. 5: Qing Period (中國人口史第五卷清時期, *Zhongguo Renkoushi Diwujuan Qingshiqi*) by Cao (2001).

## 3. Methods
### 3.1 Time series and cross checking of the variables

We used the retrieved records to build several time series for different variables and performed tests to cross check the reliability and robustness of the drought data. First of all, we gave strict definitions for drought variable. In this analysis, the drought variable only corresponds to those records with the vocabulary category of drought (such as 天旱 *Tien Han*、亢旱 *Kang Han*、苦旱 *Ku Han*, all with category code 3001), enduring scorching sun, sultry weather (e.g. 恆陽 *Heng Yang*、恒暘 *Heng Yang,* all with code 3002), dried out of different types of water body that includes dried watercourses (e.g. 水竭 *Shuei Jyue*, code 3011), dried tide/sea water, (e.g. 潮水涸 *Chao Shuei He*, code 3021), dried lake/pond (e.g. 湖涸 *Hu He*, code 3031), dried underground water (e.g. 無泉 *Wu Quan*, code 3041) and dried river/creek (e.g. 河水絕 *He Shuei Jyue*, code 3051 ). We further divide the drought records into two groups; one group that consists of pure drought vocabularies (code 3001 and 3002) is interpreted as meteorological drought, and the other that consists of dried water bodies (code 3011-3051) is interpreted as hydrological drought. Figure 2 shows the comparison of the time series. We found that higher number of hydrological drought records often occurred in the years having more meteorological drought records. Correlation coefficient of the two is 0.67, and the coefficient increases to 0.77 when only severe drought (to be explained later) is considered. This finding indicates that the large majority of hydrological drought in China can be caused by metrological drought albeit the relation is not perfect.

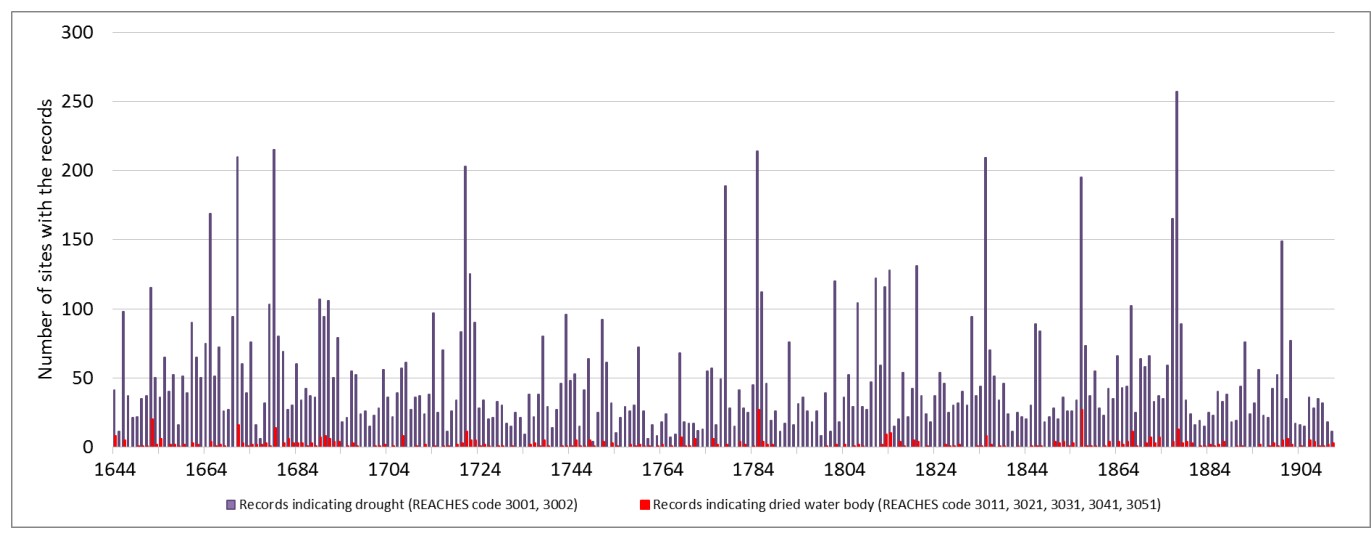

**Figure 2: Comparison of drought records that indicate pure drought vocabularies (interpreted as meteorological drought) and dried waterbody (interpreted as hydrological drought).**

To further ramify the impact of different drought types, we divide drought events into normal drought and severe drought. The criteria of severity assignment are based on the attribute of the vocabulary used and the duration described in the record. Records that contain adjectives indicating extreme or severe drought (magnitude code 2 and 5) or the event lasted for more than two months (time-duration code 6 and 7) are categorized as severe drought. In contrast, those records that do not have any adjectives indicating the severity or those with duration less than two months are categorized as normal drought. In this way we form the annual drought population and annual severe drought population series and Fig. 3 shows the time series plot of these two series. It is obvious that severe drought and drought series have good consistency and match very well especially for the peak years of the drought events. Remarkably, those peak drought years mostly appeared in the mid to late 17$^{th}$ century and after late 18$^{th}$ century. The top 3% of the drought years (those with more than 169 records of the year) are the years of 1665, 1671, 1679, 1721, 1778, 1785, 1835, 1856 and 1876.

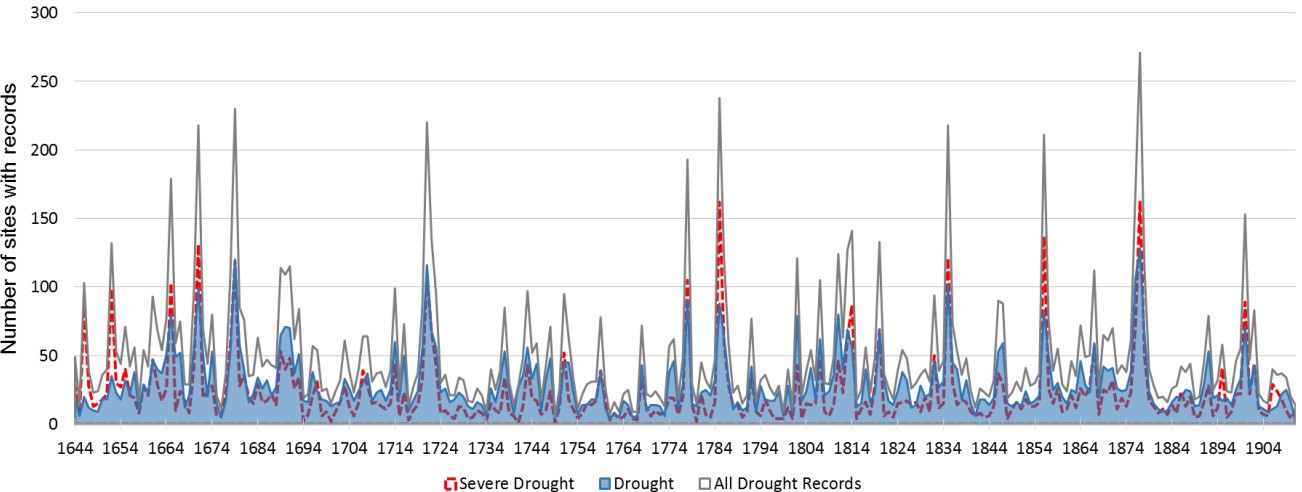

**Figure 3: Time series plot of annual normal drought, severe drought, and all drought populations in REACHES during the Qing dynasty.**

## 3.2 Spatial analysis of Kernel density estimation

Precise geographical information in REACHES makes it relatively easy to perform spatiotemporal analysis of the climate data series derived from it. There are in total 1,660 sites representing cities, counties, or prefectures in the REACHES database. We performed the spatial analysis of kernel density function for the drought data. We first determine the number of certain kinds of records each year and the numbers are summed over the time period defined for the analysis. For example, in the 1720-1740 severe drought, Baoshan district of Shanghai City had three drought records occurred separately in the years of 1720, 1723 and 1724. The frequency for the period of Baoshan district would therefore be three. We then use ArcGIS (version 10.4) to relate the data to the maps and implemented kernel density function to conduct spatial analysis.

Kernel density estimation is a non-parametric method to estimate the probability density function of a random variable. In ArcGIS it is often used to calculate density of point features around each raster cell depending on the geographic distance and value of certain features. Namely, every cell value is the highest at the location of the point features and diminishes with increasing distance from the point and reaches zero at the search radius distance, called bandwidth, from the point. On every data point, a kernel function K can be expressed as Eq. (1):

$$\hat{f}(x,y) = \frac{1}{nh^2} \sum_{i=1}^{n} K\left(\frac{d_i(x,y)}{h}\right) \tag{1}$$

where $\hat{f}(x,y)$ is the estimated density value at location $(x,y)$, $n$ is the value of the point of concern (for example in the case, $n$ is the number of drought records during the period), $h$ is a measure of bandwidth (for a circular kernel it is the radius of the circle), $d_i(x,y)$ is the distance between the point $i$ and location $(x,y)$. Eq. (1) shows that $K$ is a density function characterizing how the contribution of point $i$ varies as a function of $d_i(x,y)$.

Thus, the method estimates a smoothly curved kernel surface fitted over each point depending on the point value and values of its neighboring points within desired bandwidth. The choice of bandwidth is therefore important since larger bandwidth would result in more smoothened surface by considering point value of larger distance. In the environment of ArcGIS, the kernel density estimation is based on the quadratic kernel function (Silverman, 1986).

## 3.3 Social network analysis

To explore statistical characteristics of the relations between one variable and other variables in the drought data set, we performed social network analysis. This analysis affords us to understand more quantitatively the underlying relations of all variables. Social network is a statistical method widely applied in sociology to explore how different agents are connected with each other so as to decide their distinctive network types and to analyze if and how various types of networks would influence individual behaviors and performances (Lin, 1999; Scott, 2017). In this study, we treated each variable as an independent agent and then calculated their statistical relations.

Generally, social network analysis adopts a pairwise approach to calculate relations between variables (e.g. if agent A is linked with agent B). Thus, to implement the analysis, data transformation is needed to display and inventory events (with specific codes) under every single record. For example, a record says that 'in 1833, Guangling county of Shanxi Province, the misery [of the year] began with drought, followed by excessive rainfall, and ended with frost damages. The rice costed a thousand per dou (tradition Chinese volume unit, equivalent to 10 Chinese liters). There hadn't been so extreme in the previous several decades. ' (1833 年。山西省廣靈縣。始苦於旱，繼潦於雨，終隕於霜，斗米千錢，數十年來未有如是之極也)(record ID 2945-25). To perform the social network analysis, we decomposed the record so that the drought, excessive rainfall and frost damages each is listed as a separate event, and then we calculated the pairwise coefficients between different variable pairs, e.g., drought with rainfall, drought with frost, or drought with rice price, etc. When all events of every record have been decomposed and restructured following the algorithm, then the analysis could be performed to estimate the magnitude of every event (i.e. variable) and to calculate the strength of connections among those variables during certain time period. The function of edge list was applied to perform this analysis under Gephi software (https://gephi.org/, last access 15 March 2020).

## 4. Results

### 4.1 Chronologies of drought and related variables

Figure 4 shows the time series of all variables (in their annual values) reconstructed from the REACHES database. There are two temperature time series in the lowest part of the figure for comparison purpose to be discussed later. Naturally there are multidecadal and centennial variabilities which are more obvious in some periods amid regular fluctuations in 1644 - 1911.

One of the unique features of Fig. 4 is the inclusion of the locust outbreak series. While there had been studies on locust outbreak and climate, Wang (1985) was perhaps the first one to utilize Chinese historical records of locust outbreaks to investigate their temporal and spatial behaviors. The analysis we present here will further reveal how close this biological phenomenon is related to the climate factor.

We first observe that there appears to be a good synchronization of large amplitude events (high peaks in the curves) for all variable in six periods highlighted by light brown columns. These periods are roughly: 1665-1680, ~1720, 1770-1790, 1830s, 1850s, and 1870s, and the periods are also correspondent to the top 3% drought years in the previous analysis (Sect. 3.1). The most prominent of the synchronization is between the locust and drought series, that is, locust outbreaks occurred most often during high drought occurrence periods. The close relation between locust outbreak and drought has been known in China since ancient time. For example, the famous poet of Northern Song dynasty (960-1127 AD) Su Shi (蘇軾,1037-1101) wrote in his poem 《Rhymed after Zhang Chuandao's poem Happy for the Rain, made after I prayed for rain in Changshan Temple》: "As always locust and drought will occur together, this is what I learned from old farmers". Chen Fangsheng（陳芳生）, a Qing dynasty scholar who authored a book *A Study on Catching Locusts*《Bu Huang Kao 捕蝗考》published in 1776 in which he commented "when drought

becomes extreme, locust outbreak occurs". The synchronization in the two series thus provides a statistical verification of this ancient wisdom.

The synchronization is not limited to drought and locust. Other variables exhibit similar behavior. Thus Fig. 4 shows that when drought occurs, it is very likely that locust outbreak will occur as well as crop failure, famine and socioeconomic turmoil. We have not yet gone into detailed ramification on the cause and effect analysis of this synchronization, but the order of events as described in the previous sentence seems to match "common sense" that under the general background of drought climate, locust outbreak has a high chance to occur which, in turn, results in crop failure causing famine. And finally, in the absence of effective intervention of the government, socioeconomic disasters occurred. We are currently collecting more evidence to further study such links.

In order to further ramify this synchronization feature, we calculated the 9-years running variance for the time series as shown in Fig. 5. The six periods identified previous now become even clearer. Notwithstanding of the synchronization of high occurrence, however, the amplitudes of the variables are not related linearly. While the amplitudes of drought are more or less even in the six periods, the amplitudes of other variables seem to vary significantly. For example, locust outbreak was more numerous in 1660-1680, 1830s and 1850s (the highest peak). It is fewer in the rest three periods. On the other hand, there are fewer famine in the 1600s but become more numerous in the 1770-1790, 1850s and peaked in the 1870s. Socioeconomic turmoil has relatively low amplitude in the early and mid-Qing dynasty until 1820s when it shows an increase and peaked in 1870s. This is just to say that the correlations of these series are not perfect and their relations are nonlinear albeit they have a tendency to occur in the same period. Table 2 shows correlation coefficients among all those variables. We see that drought is highly correlated with crop failure and famine but is less correlated with locust and socioeconomic turmoil. This probably can be interpreted as that drought resulted directly in the crop failure which very likely led to famine, but there were circumstances that locust outbreak did not occur during the drought. The real reason needs to be investigated further. It is possible that measures of locust prevention had been taken sometimes effectively and not so in other times. Likewise, the socioeconomic situation is not that directly linked to drought if the government provided adequate relief (e.g., provide food, tax waiver, etc.). We plan to look into more details about the relations among these variables in the future.

To further understand the multidecadal and centennial variabilities, we compare these series with two temperature series: one reconstructed from REACHES (Lin et al., 2019) and the other North Hemisphere ensemble temperature from Frank et al. (2010). The two temperature series show similar trends in anomaly which appears to augment the reliability of data derived from REACHES. Figure 4 shows that higher frequency of all the drought-related variables tend to occur more often when temperature is lower in the 17[th] century and after late18[th] century. Thus colder periods in historical time also tend to be periods of low agricultural yields and social instability, which is consistent with some previous studies (Pei et al., 2019; Su et al., 2016; Wei et al., 2017)

369

370

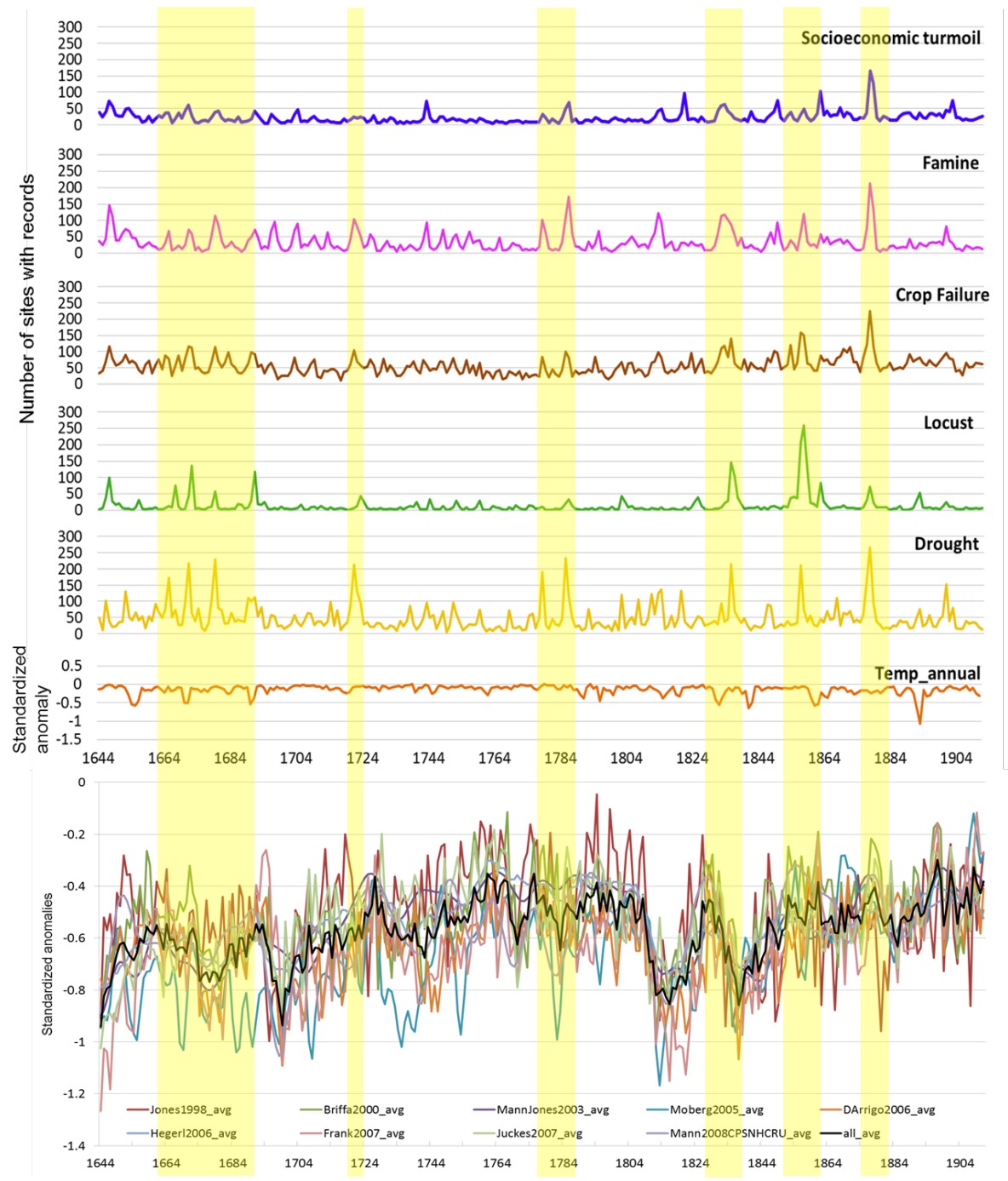

**Figure 4: Time series plots of socioeconomic turmoil, famine, crop failure, locust, drought, and monsoonal China annual temperature anomaly reconstructed from the REACHES. The bottom panel shows ensemble North Hemisphere temperature anomalies from Frank et al. 2010. Periods with more frequent drought records are highlighted in yellow bars.**

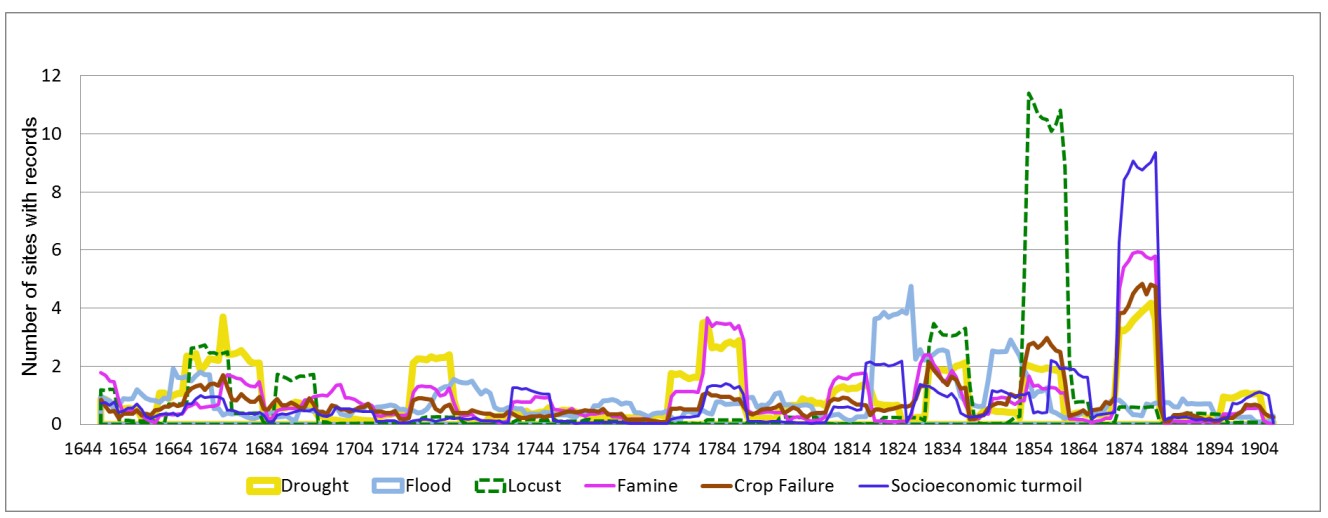

**Figure 5: The 9-Years running variance for the multiple variables of the REACHES.**

**Table 2: Correlation coefficients of REACHES multiple variables**

| Drought Category | Variable | Drought | Locust | Socioeconomic Turmoil | Famine | Crop Failure |
|---|---|---|---|---|---|---|
| Meteorology & Hydrology | Drought | 1.000 | | | | |
| Ecological-related | Locust | 0.375 | 1.000 | | | |
| Socioeconomics | Socioeconomic turmoil | 0.442 | 0.287 | 1.000 | | |
| | Famine | 0.618 | 0.379 | 0.718 | 1.000 | |
| Agriculture | Crop Failure | 0.626 | 0.511 | 0.685 | 0.675 | 1.000 |

All coefficients have P-value <0.001. High correlation coefficients are shaded with dark gray and medium correlation coefficients are shaded with light gray. Note that we have in Sect. 3.1 identified the correlation coefficient between meteorological drought and hydrological drought as 0.67 which increases to 0.77 when severe drought is considered.

**4.2 The spatial pattern of drought during the six high occurrence periods**

In the above analysis, we identified six periods in which the drought-related variables exhibit nearly synchronous fluctuations. We now would like to demarcate more precisely these periods based primarily on the drought series. In doing so, we also considered auxiliary information we found in the aforementioned historical documents such as Draft History of the Qing, Actual Veritable Records of Emperors of the Qing Dynasty and the Gazetteers. For example, during the period of 1720-1740, the severe drought mainly occurred around 1720 but the drought soon continued in 1730s and 1740s, and their socioeconomic effects were closely linked, so it seems to be strategically prudent to consider the period 1720-1740 as a high severity period. On the other hand, the three severe drought periods in the 1800s appear to be less closely linked in time (one peak every two decades) and the auxiliary information show that their socioeconomic consequences were not closely linked either. Thus we do not see the necessity of lumping the

three periods together. In this way, we demarcate the six severe drought periods in Qing dynasty as follows: 1665-
1691, 1720-1740, 1770-1790, 1830-1850, 1850-1870, and 1870-1890.

In the following, we will examine the spatial distributions of the drought and the corresponding distribution of locust
outbreaks and socioeconomic turmoil. We will also provide a summary of the population and war frequency and the
results of social network analysis in these six periods.

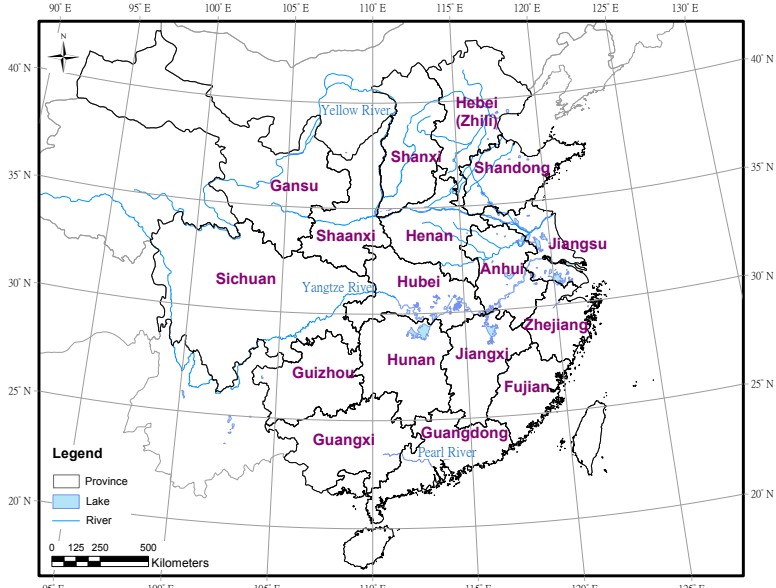


**Figure 6: Provincial boundaries and names in eastern part of China. The basic latitude and longitude grid map**
**is from ESRI data set, provincial and city/county boundary and river shapefiles are from China National**
**Bureau of Mapping and Surveying, Digital Map Database of China (1990 version).**

The following discussions will frequently mention the provincial names in China and hence may be confusing for
readers not familiar with Chinese geography. Figure 6 is a simplified map showing the locations of the provinces
mentioned in the discussion for reference.

### 4.2.1 Period 1665-1691

Figure 7 shows the overall situation of drought-related variables in the period 1665-1691 including the spatial distribution of the kernel density function of drought (7(a)) and associated locust outbreaks and social turmoil (7(b)), and the social network chart (7(c)). Figure 7(a) shows the drought kernel density distribution which is an indicator of the severity of drought. It is seen that in this period drought was fairly wide spread in the eastern half of China from as far north as Hebei province and as far south as Guangdong province. There appear to be two "epicenters" of drought, one in the lower Yellow River flood plain (Hebei and Shandong) and the other in the middle and lower reaches of Yangtze River (Anhui and northern Jiangsu). There are also secondary centers in Hubei, Jiangxi, and Zhejiang. The drought in the south such as Fujian and Guangdong were relatively mild.

Here we would like to stress an important caution when reading the spatial patterns: this should be viewed as a *map of anomaly* instead of a map of humidity. This is because the drought index (as well as other severity indices such as temperature) derived here represents the *deviation from the local norm*, and the local norm can be quite different in different locations in a country as large as China. Thus, a certain dry condition lasted for, say, a month may well be considered a drought in the normally humid Southern China but could be regarded as a normal condition in the Loess Plateau region in Northern China because their norms are quite different. There is a need of an algorithm that converts the anomaly index to actual humidity which must consider this local norm factor. An investigation is being conducted on this subject.

Figure 7(b) shows the corresponding spatial patterns of locust outbreaks and social turmoil in this period. We observe that the patterns are both similar to that of drought and both are also centered around lower Yellow and mid- and lower Yangtze, which again vindicates the ancient wisdom of the close relationship between drought and locust (and the consequential social turmoil they might cause).

The period of 1665-1691 is about two decades after the collapse of Ming Empire in 1644 which was replaced by the Manchurian Qing Empire after the bloody wars among different parties including Ming, Qing and the rebellious farmers led by Li Zicheng (李自成). Population of China was substantially reduced due to the wars. But even at this time, revolts in different parts of China were still going on. Severe drought and locust outbreaks occurred in 1670-1672 and appeared again in 1679 and 1689-1691. Given the very similar patterns of drought, locust outbreak and social turmoil distributions at the time, it is possible that many of these revolts were motivated by the food shortage (thus climate-related) in addition to those motivated by political considerations (e.g., The Revolt of the Three Feudatories in 1673-1681). This will require further studies to ramify.

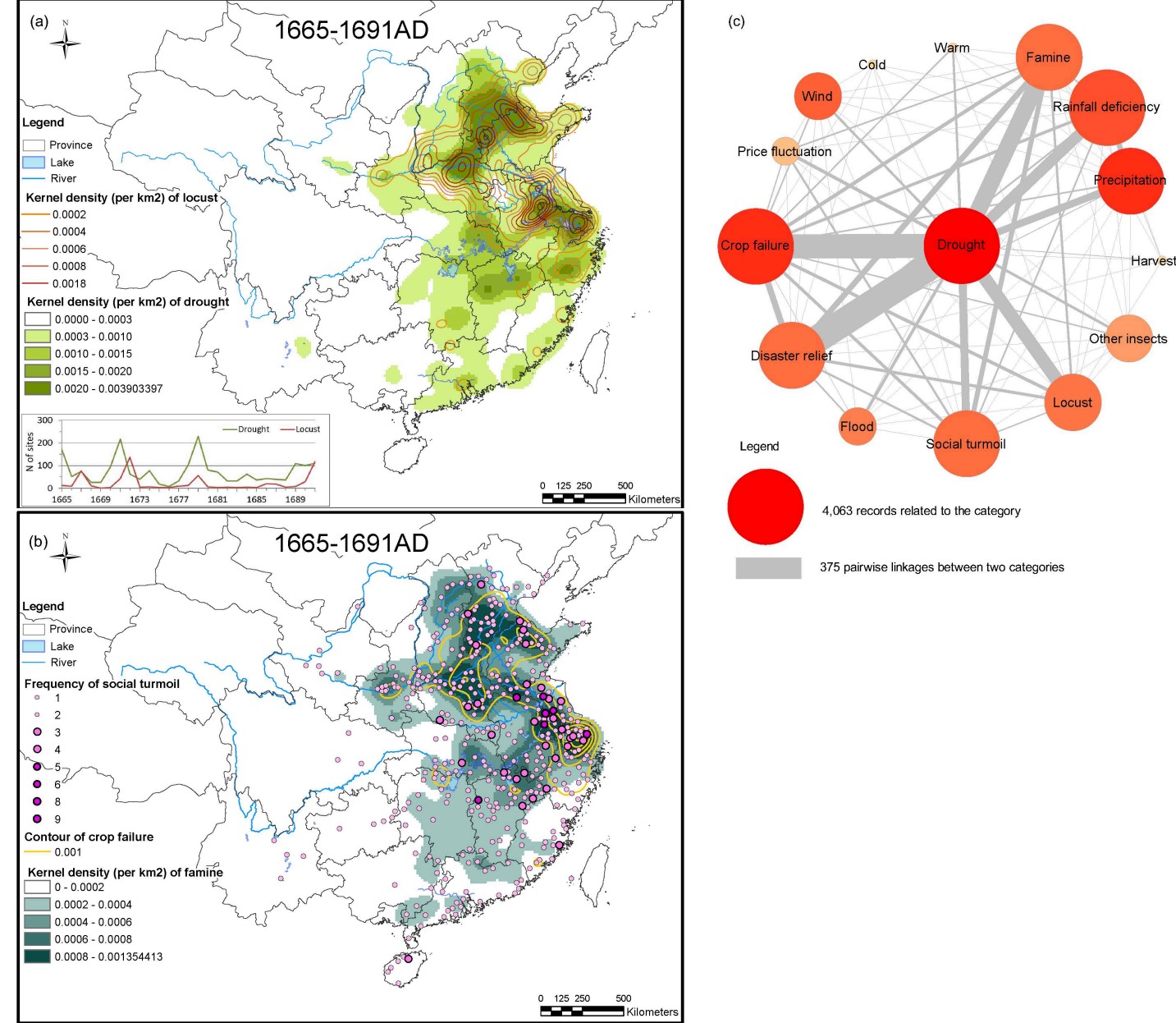

**Figure 7: Spatial distribution of the kernel density function of drought (a) and locust outbreaks and social turmoil (b), and the social network chart (c) in the drought period 1665-1691.**

### 4.2.2 Period 1720-1740

The 1720-1740 drought as shown in Fig. 8(a) appears to be not as wide spread as that in 1665-1691. While there are still two centers: one in Lower Yellow and another in Lower Yangtze, the latter appears to be the more severe one. The corresponding locust breakouts and social turmoil are also centered in these two regions although that in the Lower Yellow appears to be somewhat more extensive. There were locust outbreaks reported in Fujian, Guangdong and Guangxi provinces but there wasn't much corresponding drought reported in these areas. There was social turmoil

in areas not affected by drought or locust also. Were these disturbances not climate-related? The reason is unclear and
will requires further studies.

There is a term in Chinese historical community called High Qing era or the Kang-Yong-Qian Prosperity Period (康
雍乾盛世)，referring to the period reigned by the Emperors Kangxi, Yongzheng and Qianlong when Chinese economy
was relatively prosperous and is often considered as the golden era of the whole Qing dynasty. The demarcations of
the period differ among scholars but a generally usable one is 1684-1799 (Guo, 2002). The period of 1720-1740 of
concern here falls in this golden era.

Emperor Kangxi died in 1721 and was succeeded by his son Emperor Yongzheng after a conspiratorial process. But
Yongzheng himself died in 1735 after another process also full of rumors and his heir was Emperor Qianlong who
died in 1799. Thus Emperor Yongzheng was the main ruler in the period 1720-1740. During this time, the whole
China empire was generally stable. There were some wars with minority tribes such as the one with the Mongolian
Goldan Tseren of Dzhungar Khanate but not extensive.

Population of China had grown general during this period, but the drought and locust outbreaks certainly left their
marks in the affected areas. According to a 1725 population record, Shandong's population grew slowly at this time
and the population of Jiangsu declined. Later famine occurred in the north of Huai River flood plain around 1738
which is drought-related and lasted for several years (Fig. 8(b)).

The social network analysis (Fig. 8(c)) showed some interesting feature, that is, the links between the pairs of drought-
locust, drought-crop failure, and drought-famine, though still significant compared to other pairs, were much weaker
than in the previous period. This seems to indicate that while drought occurred, the societal impact was relatively mild.
Although the real reasons still need to be researched carefully, there are documentary evidence showing that the
governmental policies, such as shipment of grains to and tax relief for the disaster-affected areas, and the
implementation of locust elimination, might have played a significant role here.

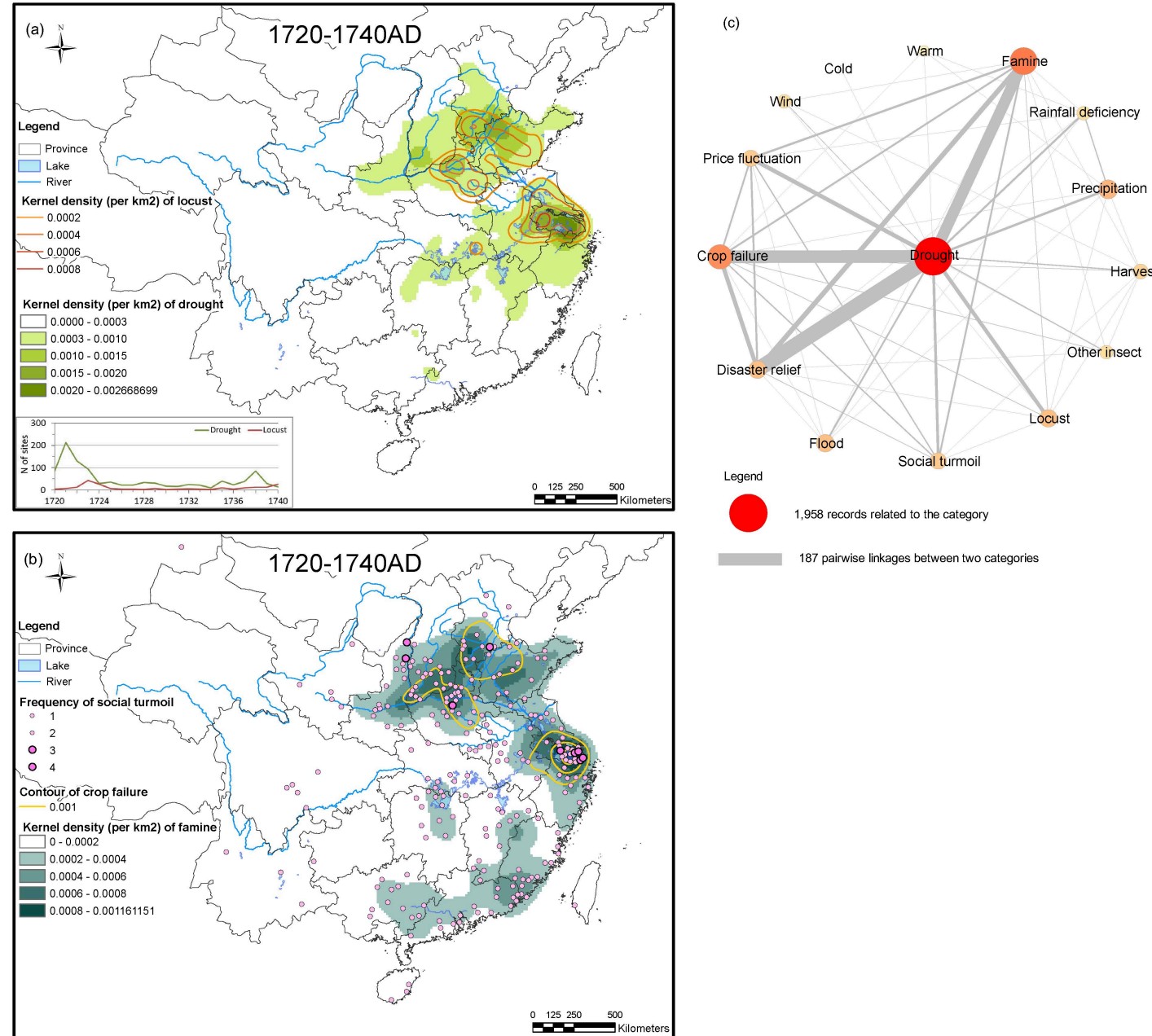


Fig. 8 Spatial distribution of the kernel density function of drought (a) and locust outbreaks and social turmoil (b),
and the social network chart (c) in the drought period 1720-1740.

### 4.2.3  Period 1770-1790



This period was entirely under the reign of Emperor Qianlong. Five years later in 1795 he claimed to pass the throne
to his son Emperor Jiaqing（嘉慶）although in reality he was still in control until he died in 1799. China was relatively
stable politically in this period. It was reported that the national population had returned to that in Ming dynasty Only
some minor revolts occurred in Western China in 1771-1776 by Jinchuan tribes （大小金川）who were related to
Tibetan, and in Taiwan in 1786-1788 led by Lin Shuanwen （林爽文）of Tiandihui (天地會，literally Heaven and
Earth Society, a secret society that still exists today). A British diplomatic mission led by George Macartney was
granted an audience with Emperor Qianlong in 1793, an event that became the prelude of wars between China and
Western Powers later.
The spatial distributions of drought and the corresponding locust outbreaks and social turmoil as shown in Fig. 9(a)
and (b) are both similar to that in 1720-1740, namely, both are still centered on Lower Yellow and Lower Yangtze
regions and a smaller center in Guangdong in the south. Drought appears to be more mild but affected areas appear to
be wider, but social turmoil appears to be somewhat worse than in 1720-1740.
The social turmoil was likely inked to social inequality due to the gap between the rich and poor. The wide spread of
locust outbreaks also prompted the government to pay more attention to the technology of locust eradication and the
previously mentioned *A Study on Catching Locusts* was published in this period. Drought in this period led to a
reduction in crop yield resulting in famine and sharp rise of grain price around 1779 and 1786. Figure 9(c) shows the
social network analysis of this period.

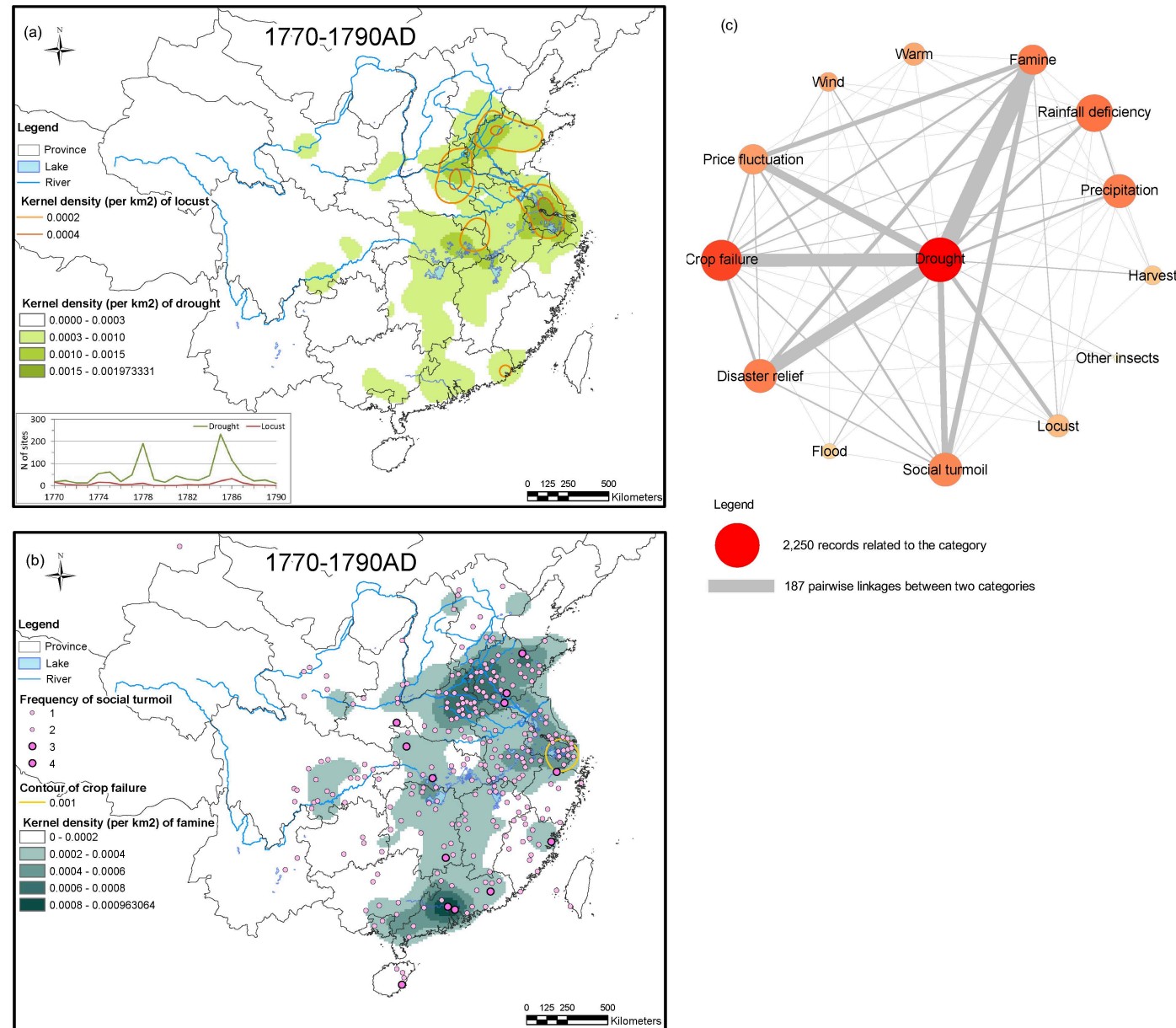

**Figure 9: Spatial distribution of the kernel density function of drought (a) and locust outbreaks and social turmoil (b), and the social network chart (c) in the drought period 1770-1790.**

### 4.2.4 1830-1850 period

The overall political and economic situation of China was going downhill after nearly 200 years of Qing rule. The middle age of Emperor Qianlong's rule was generally considered as the peak of heyday of Qing dynasty and by 1830 China was facing increasing economic downturn after Emperor Daoguang（道光）assumed the throne in 1820 when his father Jiaqing died. Many Chinese historians consider that the overspending by Qianlong is to blame for the declining economy, especially from the discovery of the stunning corruption by his most favored minion, Heshen （和珅）who was believed to have amassed illegally a net wealth equal to fifteen years of total tax collection by the Qing government. But from today's point of view, the challenge from Western Powers at the time such as the British, French

and German was probably a more serious problem. Although there were some smaller riots occurring in this period, the most well-known armed conflict was the Opium war occurred in 1840 between Qing and British Empires. After the war, the inability of Qing government to cope with the international pressure in various aspects was completely exposed and ushered in more serious challenges internally.

But perhaps part of the blame can be attributed to the climate factor. Figure 10(a) and (b) show the drought and the corresponding locust outbreaks/social turmoil spatial patterns. The drought condition looks similar to that in Fig. 8(a) but the locust/social turmoil looks much worse and more wide spread than in 1770-1790. Locust outbreaks reached as west as Shaanxi province in Northern China and Sichuan province in Central China which were largely spared in the previous period, and as far south as Guangdong and Guangxi provinces. Hubei province in Central China was plagued badly by locusts and social unrest. Even the relatively wealthy province of Zhejian had seen serious locust outbreaks and social unrest. Such relations among drought-related factors can also be seen in the social network analysis in Fig. 10 (c). But overall, drought is mostly related to crop failure and famine. Drought in this period did not seem to link strongly with precipitation and the reason is unclear at the moment.

An interesting feature worthy of note here is the spatial patterns of locust outbreaks and social turmoil which exhibit an east-west (E-W) orientation. There are apparently two prominent E-W belts: the north one along Shandong, southern Hebei and southern Shaanxi provinces; the south one along Jiangsu, southern Hubei and northern Hunan provinces.

Documentary sources show that the national population continued to grow rapidly but per capital arable land declined significantly. Whether or not the decrease in arable land was due to climate factor remains to be studied, but there is little doubt that the society was under great stress of resource competition. This eventually led to numerous uprisings.

One would expect that many of the uprisings must be due to food shortage given the declining production of grains. Curiously, the rice price information in 1825-1860 we obtained for several places including Suzhou (of Jiangsu province), Hubei and Guangxi paints a fairly different scene. Suzhou's rice price fluctuated more wildly but appears to be steady when averaged throughout the period. Guangxi's rice price was very steady and Hubei's price was even declining during this period. Perhaps this phenomenon only pertains to these few cities as we have not studied the situation of other locations. This will be left for future research.

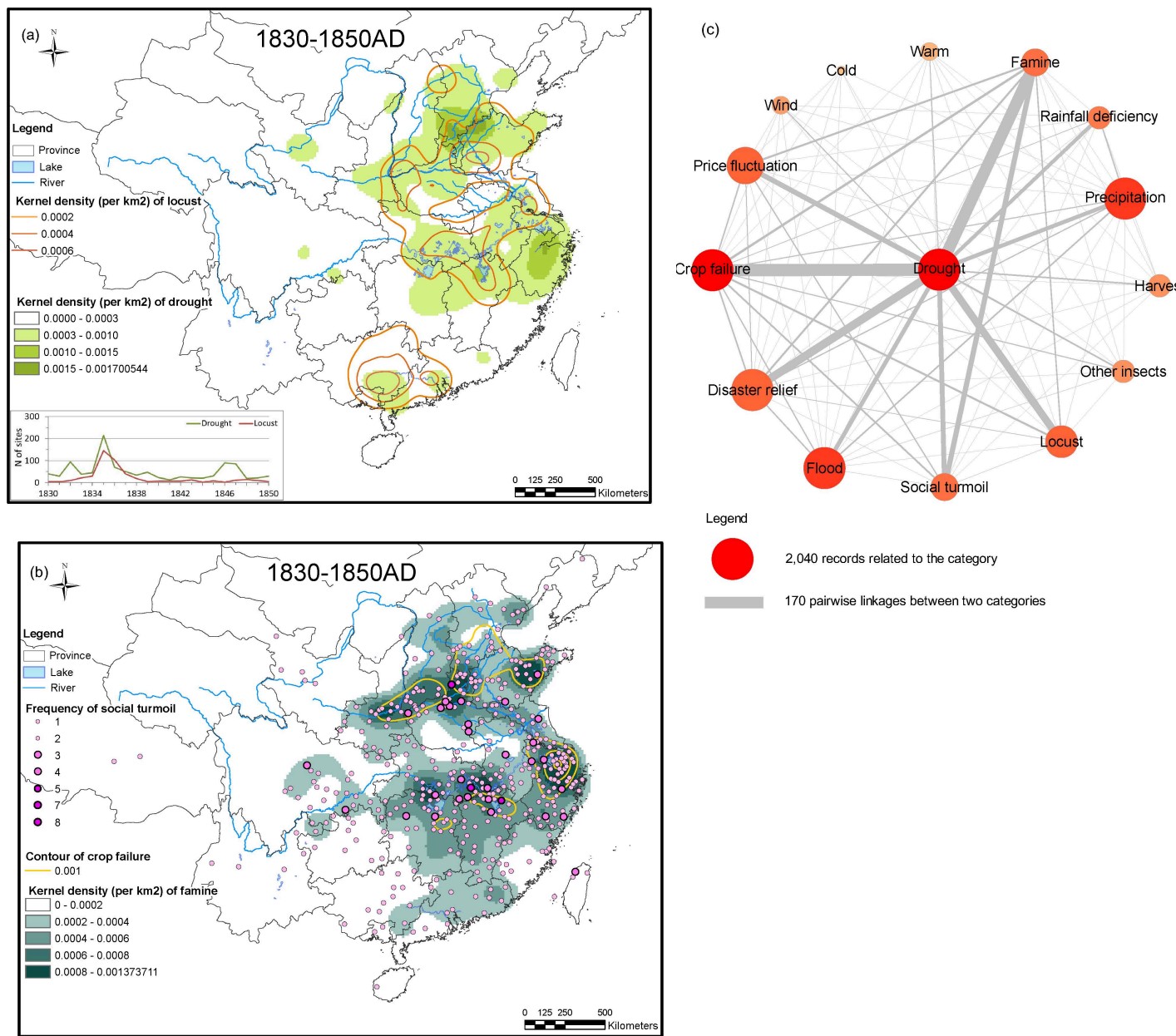

**Figure 10: Spatial distribution of the kernel density function of drought (a) and locust outbreaks and social turmoil (b), and the social network chart (c) in the drought period 1830-1850.**

### 4.2.5 Period 1850-1870

In this period, drought condition in the Yellow plain in Hebei and Shandong in Northern China worsened and locust outbreaks were also wide spread similar to the 1830-1850 period. Fig. 4 shows that locust outbreak frequency in this period is the highest in the whole Qing dynasty. Famine occurred in locations where locust outbreaks were severe. The center of the worst condition – severe drought, heavy locust outbreaks and high famine density – was in the combined region of Hebei and Shandong provinces forming a northwest-southeast (NW-SE) belt as shown in Fig. 11 (a) and (b). This is in contrast to the E-W orientation of the corresponding distributions in the previous period as

shown in Fig. 10. Could this difference in orientations be an indication of the regional atmospheric circulation pattern change?We will conduct further studies on this subject in the future.

This was a time that political situation of Qing became increasingly unstable. Emperor Daoguang dies in 1850 and was succeeded by his son Emperor Xianfeng (咸豐). Rebellions erupted all over China and the deadliest one was the Taiping Rebellion (太平天國之亂) led by Hong Xiuquan (洪秀全), who was a self-proclaimed Christian and who claimed himself to be the brother of Jesus Christ, that started in 1850 and ended in 1864. This is thought to be the largest scale civil war in China in Qing dynasty and the bloodiest. Casualties was estimated in the range of 10-30 million and another 30 million fled to other and foreign settlements of China. This 14-year war impacted 18 provinces but the region south of Yangtze River suffered the most casualties in this conflict.

In the Central and Northern China, it was another bloody war, the Nian Rebellion (捻亂), that caused severe damages to the country. "Nian" was one of the cult-like secret societies active in Central China especially along the region between Anhui and Henan provinces after the disastrous Yellow River flood in 1851 that deluged hundreds of thousands of square miles killing a large number of people. The scale of the societal turmoil was not as large as Taiping but still caused serious social disruptions and loss of lives. The rebellion lasted from 1853 to 1868. There is no doubt that this and Taiping Rebellions caused huge damages to Chinese economy and contributed to the many heavy turmoil circles in Fig. 11 (b). Since these turmoil circles are unrelated to drought, they could appear in locations not affected by drought. The social network analysis in Fig. 11(c) shows the strong links between drought-locust, drought-crop failure and drought-famine.

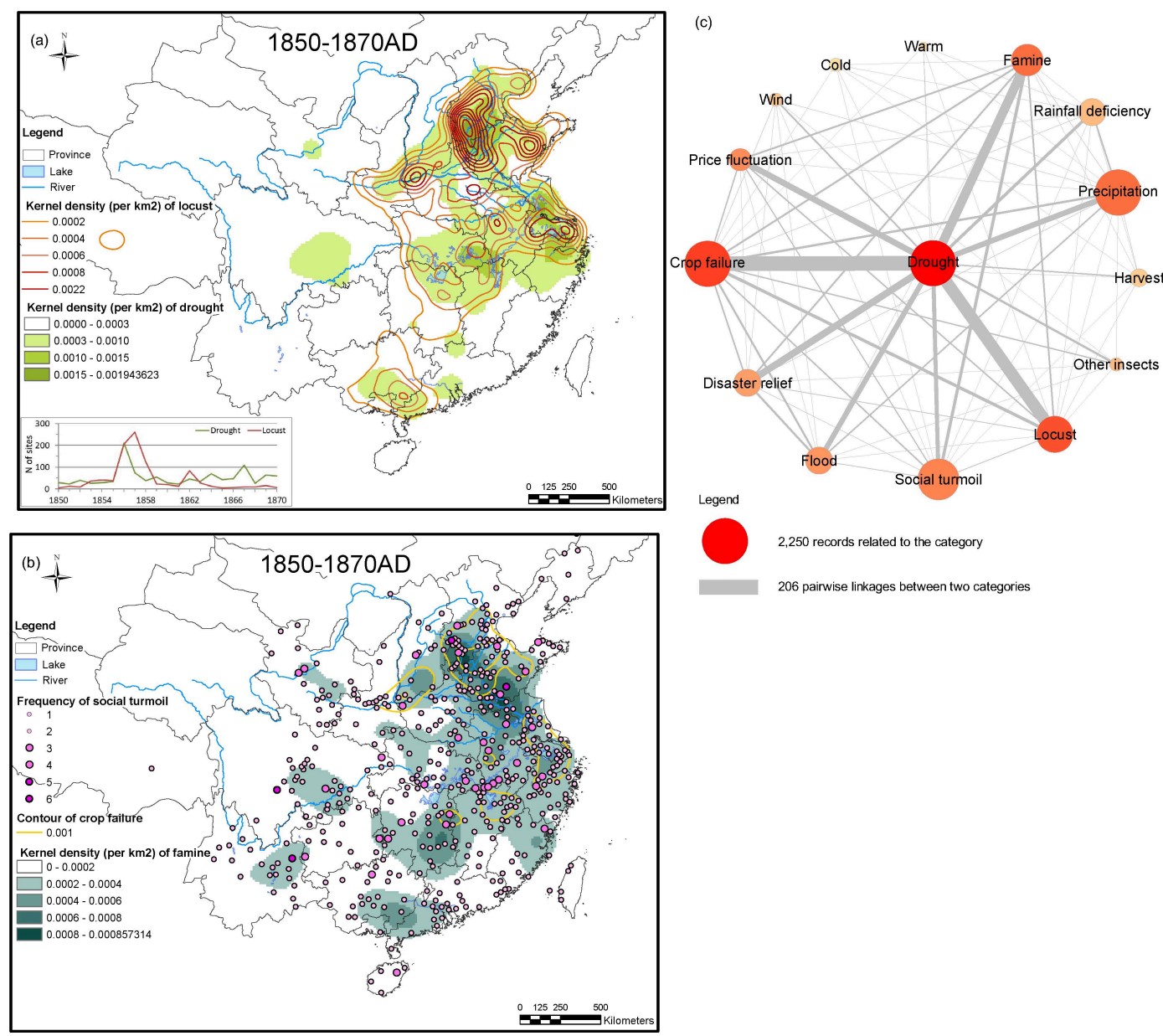

581

**Fig. 11 Spatial distribution of the kernel density function of drought (a) and locust outbreaks and social turmoil**

**(b), and the social network chart (c) in the drought period 1850-1870.**

584

### 4.2.6 Period 1870-1890

586

Figure 12 (a) and (b) show the drought spatial distribution and the corresponding locust outbreaks/social turmoil distribution. In the Northern China, the two spatial patterns match quite well. The drought was centered on the region between Hebei and Shandong. The center of locust outbreaks/social turmoil was larger encompassing Shandong, Hebei, Shanxi, and Shaanxi. There was a small pocket of drought as well as locust/social turmoil in Sichuan province. The whole belt shows a NE-SW orientation. There was also a small center of drought and locust/social turmoil in northern Jiangsu. The rest of China, especially the south, appeared to be under no serious threat of drought although there were still sporadic social disturbances occurred in various places.


This period was near the end of Qing dynasty which lasted 21 more years and finally ended in 1911. China was
devastated by the Taiping Rebellion in the previous period and political situation was highly unstable as foreign
powers posed huge challenges to Qing government. Emperor Xianfeng died in 1874 and Emperor Guangxu (光緒), a
nephew of Xianfeng but adopted as a son later, succeeded the throne although the person in real power was the
Empress Dowager Cixi (慈禧太后) who declared war to all foreign powers after the Boxer Rebellion erupted in 1900
leading to the invasion of Beijing by the allied armies of eight countries.

The social network chart in Fig. 12 (c) shows a feature unique in this period that is not prominent in other periods: the
link between famine and social turmoil is very strong (the grey bar linking the two is very thick). This can also be
understood from Fig. 5 that the numbers of crop failure, socioeconomic turmoil and famine are all very high in this
period and they are highly correlated. This is also possibly an indicator that the government relief effort was not
effective in this period. This is not so in other earlier periods where the grey bars linking famine and socioeconomic
turmoil are rather thin. For example, the purple socioeconomic turmoil circles in the period 1665-1691 in Northern
China in Fig. 7(b) are fairly numerous, yet the grey bar linking famine and socioeconomic turmoil is relatively thin,
possibly indicating that the effectiveness of the relief programs by the government at this earlier period. Of course,
the socioeconomic interactions between various factors can be very complicated and nonlinear, and the above
statements should be regarded as temporary conjectures instead of definite conclusions. Further analyses need to be
performed to fully understand such relations.

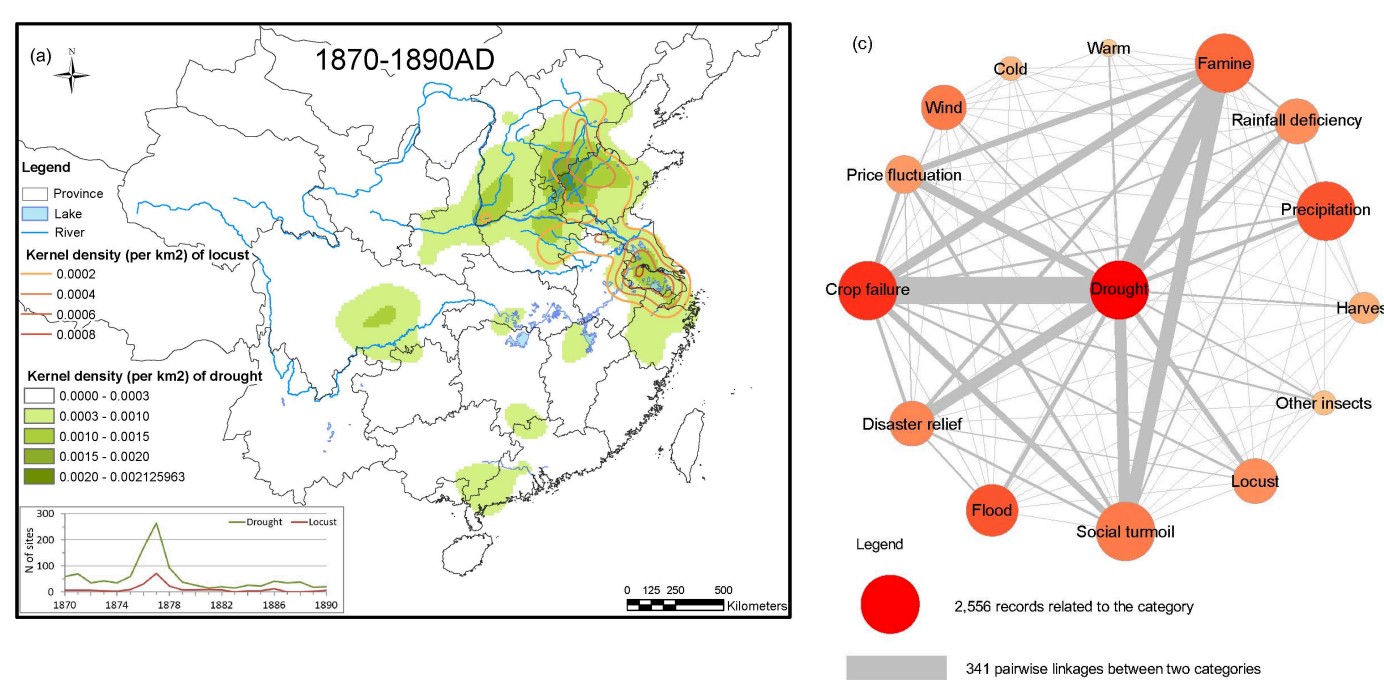


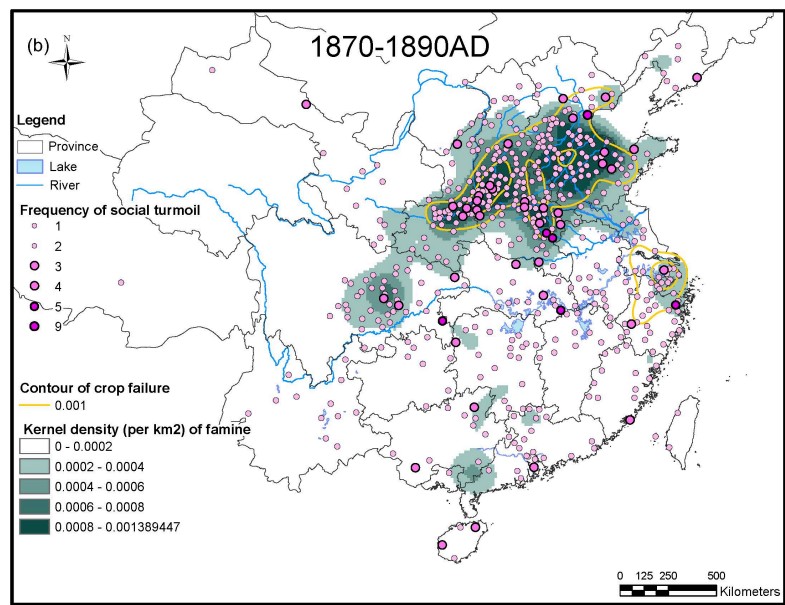

**Fig. 12 Spatial distribution of the kernel density function of drought (a) and locust outbreaks and social turmoil (b), and the social network chart (c) in the drought period 1870-1890.**

## 5. Discussions

In the above, we proposed a new scheme to define the severity of drought reported in the Chinese historical documents and the relations between drought and other related ecological, agricultural, and socioeconomic variables. Instead of building mixed criteria to evaluate drought severity by artificial and often subjective judgment on the texts of the written records and assign them into different grades, this study adopts a more objective scheme to extract information from descriptive statements for reconstructing drought chronology. Drought and its associated variables are carefully categorized individually so as to build their distinctive chronologies. The advantage of this scheme is the straightforwardness of the approach without subjective guessing of the severity. It also allows cross check among variables and thorough statistical examinations on the relations.

In this study, we examined the literal meaning of drought mentioned in the written records by considering the four categories of drought as defined by Brázdil et al., (2018) and Heim (2002). We found that in the Chinese literature context, meteorological drought and hydrological drought are closely correlated with each other (R=0.67). The hydrological drought had even stronger correlation with severe meteorological drought (R = 0.77). This is easy to understand in that severe meteorological drought must necessarily result in hydrological drought unless the society has access of highly reliable non-precipitation water resource (such as underground water), which was nonexistent in historical China. Agricultural drought denoted by crop failure in this study is also highly correlated with meteorological and hydrological drought (R = 0.63). On the other hand, socioeconomic drought as represented by famine is also correlated fairly well with meteorological and hydrological drought (R=0.62). But the correlation between socioeconomic turmoil and meteorological and drought is weaker (R=0.44) possibly because there are other causes for the turmoil.

We found that in the period of 1644-1911 drought and related phenomena including locust outbreak, crop failure, famine and socioeconomic turmoil tended to occur with lower temperature anomalies in the 17[th] century and 19[th] century. The trend of temperature anomalies derived from REACHES database is generally consistent with other studies (e.g., Frank et al., 2010; Ge et al., 2017). While some previous studies in China have pointed out that there were more droughts in the 16[th] and 17[th] century than in the 18[th] and 19[th] centuries (Song, 2000; Zheng et al., 2006; Shen et al., 2007; Yi et al., 2012; Ge et al., 2016), the resolution of their data did not allow them to identify intrinsic drought events at interannual and decadal level. In contrast, the good time resolution of data series derived from REACHES allows us to clearly identify the six severe drought periods as described in previous sections.

Another strength of data series derived from REACHES is the precise geographic information carried in them (Wang et al., 2018). While some previous studies were able to show spatial patterns of the drought and flood (CMA, 1981), they were not shown in precise locations and not carried fine resolutions. Our study facilitates the analysis of the spatial patterns of drought and other related climate variables, as illustrated by the discussions in the previous section. The spatial analysis reveals many features that cannot be easily shown by the series analysis alone. For example, one can obtain an immediately appreciation of the similarity of the drought and locust spatial distributions in several periods by simply inspecting the maps shown in the last section. It would be difficult to reach such a conclusion from the time series shown in Fig. 4 or Fig. 5 because the series simply show the total counts nationally. One would need to perform time series for each location to come to the same conclusion but that would be an extremely arduous effort.

To ensure the information shown on the maps are reliable, we delved into a great quantity of archival information and index data from independent sources in the context of social and economic aspects related to the severe drought events in these six periods. We found good consistency between the contents retrieved from the archival data and the analysis derived from the REACHES. For example, heavily impacted regions shown on the maps are very consistent to those provinces and cities or counties mentioned in the archival data.

We also performed the social network analysis to show the relative magnitudes of drought and related variables in the six periods and the strength of connections among all variable pairs. In general, the features seen from these social network charts are consistent with "common sense" at a qualitative level although the strength of the links between different variable pairs can be quite different in different periods. We tried to interpret some of these links to understand their implications, however, we stress that due to the complexity of human society it is not always easy to make straightforward interpretations. In human events, a small perturbation can sometimes trigger a huge consequence due to the high nonlinearity of social dynamics. We plan to conduct further studies to gain deeper insights on this subject.

## 6. Conclusions

The strength of the study is the transparency of the method to extract information from descriptive statements of drought records for reconstructing drought chronology. Our main purpose is to carry out such an objective scheme in

which different categories of drought and associated variables can be clearly defined and be carefully examined. All the data series developed in the study is made publicly accessible and can be widely applied in future studies of multidiscipline. The main conclusion of the study can be displayed in several ways: First, this study demonstrates the richness of the historical documentary records to be used in space-time analysis for revealing extreme events and their societal responses, as also revealed in previous studies such as Pfister (1992) and Brázdil et al. (2019). Careful encoding and interpreting of the records into separate categories relating to drought phenomena, instead of pooling them together for drought degree judgment, is substantial for further understanding drought contexts and estimating their associations. Second, in the Qing dynasty (1644-1911), we identify six drought periods and importantly our spatial and social network analysis reveal various spatial patterns and directionalities of the drought phenomena in the periods (e.g. a NW-SE belt in 1850-1870 drought period, and a NE-SW orientation in 1870-1890 period) and the societal impacts were also discrepant. The socioeconomic interactions can be very complicated and nonlinear, thus further analyses need to be performed to fully understand such relations.

Finally, as we have emphasized before, the climate series presented here are anomaly series and extreme care must be taken when interpret the characteristics of the series. For example, some researchers may interpret a period of high drought frequency as a "dry climate" period while in reality the high frequency only indicates that many "dry weather days" had occurred. This can be best understood by inspecting Fig. 5 where the flood time series is also included. We see that the flood frequency could also be quite high during a high drought frequency period and many flood cases are indeed due to excessive rainfall, and obviously one cannot interpret this as a "humid climate" period. A more appropriate interpretation would be that this is a period of high precipitation variability. The interpretation of other anomaly series should also consider the possibility of this type of bias. Lastly, it will be also imperative in future study to compare drought and other climate time series across different continents such as the comparison between East Asia and Europe, so that more insightful view can be acquired to better understand general circulation at the times.

**Data availability**

Data series reconstructed in this study will be deposited at NOAA national center for environmental information (https://www.ncdc.noaa.gov/paleo-search/study/23410). Or users can also contact the authors for more information on data series.

**Competing interests**

The authors declare that they have no conflict of interest.

**Author contribution**

KHE Lin set up the research topic, designed the research, performed the analysis and drafted the manuscript. PK Wang motivated the research, and comprehensively edited the manuscript. PL Pai helped collecting archival data and working on historical context. YS Lin helped retrieving and cleaning the data series, and implementing the analysis.


**Acknowledgments**
We thank for the support of the Ministry of Science and Technology, Taiwan (project no. MOST 108-2621-M-001-
007-MY3) and Center for Sustainability Science, Academia Sinica, Taiwan (project no. AS-105-SS-A04).

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
