# Peer review of "Historical droughts in the Qing dynasty (1644-1911) of China"

_Climate of the Past, 2019_

## Referee Comment (RC1) · Anonymous Referee #1 · 7 Nov 2019

This is an important topic in the research on past climate change. The topic also fits the journal. It cannot be accepted for publication, because there are lots of problems must be solved seriously.

First, the manuscript selected the drought and reconstruction and its impact on human society. So far, there are lots of studies in this direction. May I know the innovative points of this manuscript, in comparison with these existing findings? The authors did not make it clear.

Second, there is a big gap between their aim and their methods. In the manuscript, "our objective is to make every drought and associated variables as literally clear and operationally independent as possible." How the authors could evaluate the objective of "clear and operationally independent"? This object has not been discussed in the

later sections. Did authors achieve the aim? Please clarify.

Third, the authors are very proud of REACHES. I also read the paper introducing REACHES as shown in the reference of the manuscript. In fact, the database is from Compendium of Chinese Meteorological Records of the Last 3,000 Years (Zhang, 2013). This book is the basis for whole research and makes REACHES scientific and trustable. The authors should not over-emphasize the importance and innovation of REACHES.

Fourth, as mentioned by the authors, "To comprehensively compare and analyze drought and associated data series from the REACHES with other socioeconomic variables from independent data sources, several archival and index data were also collected for analysis." May I know the similar or different features in the records of these documents? The authors did not discuss enough to compare these sources.

Fifth, there are different categories of drought recorded in the historical documents. Why there are different records in the documents? Please clearly explain the reasons to have these different records in historical documents. Then, the readers will know rationale of these categorizations and see the importance of this research.

Sixth, I am not convinced by the Kernel method. It is common to have the missing data in the documents among different regions. If using the number of records, the results will be disturbed by the data availability condition. The results are thus not reliable at all.

Seventh, in Page 11, there are some linkages according to the one record, "the events would be decomposed and then displayed in a way that drought linked with rainfall, drought linked with frost, drought linked with rice price, rainfall linked with rice price, frost linked with rice price and so on to further calculate their pairwise coefficients." It is not persuasive to have such findings by only one record. In fact, the authors should revise the whole manuscript to review their findings. Please avoid similar problems.

[Figure]

Eighth, please check the language. There are some typos. Such as, it should be Guang Ling but not Quang Ling in Page 11.

Ninth, the language is not clear and concise enough. There are many redundant sentences in the manuscript, such as Page 2 "Studying past drought and humidity has been a long practiced subject in historical climatology and paleoclimatology". If the authors still want to keep these redundant sentences, why not add the references?

In terms of reference, the authors have many judgements without the proofs from their articles. For example, Page 3 "Yet, tree ring reconstruction usually suffers from growing seasonality of trees and blurred interpretation of isotopes." This is not your findings from the manuscript. There are many similar examples in the manuscript to show the authors are not careful enough to conduct the research and claim their findings.

---

## Referee Comment (RC2) · Anonymous Referee #2 · 12 Dec 2019

GENERAL COMMENTS This article discusses historical droughts and the role of human interventions in the Qing Dynasty (1644-1911) of China based on the REACHES database, which was created using the "Compendium of Meteorological Records of China in the Last 3000 Years". The main purpose of this article would be an analysis of long-term variations of droughts and their impacts on human society in China during 1644-1911. Although the methodology used might be somewhat innovative, the results were not so new and interesting as compared with a lot of previous similar papers analyzing the changes in climate and natural disasters in China during the historical period. Also, another problem of this article is that four authors of this article are the same as those of the main referenced paper by Wang,P.K. et al.(2018) which introduced the construction of the REACHES database. For most of the readers of

this article, the REACHES database might be unfamiliar and unrecognized. Therefore, authors should explain the REACHES database in detail at first using visual diagrams and charts, such as shown in Wang,P.K. et al.(2018). However, this article would be acceptable for publication after minor revisions.

SPECIFIC COMMENTS I. Introduction

P.2 Line 26: Studying past drought and humidity » Studying past drought and flood After this sentence, previous related papers should be referred to. Line 34: documented records » historical documents (e.g., ****,*****)

P.3 Line 30-31: Three-, five-, ———- is the most commonly practiced method (previous articles should be referred here), so that ———

2.Data

P.5 Line 22: The data source of this study mainly comes from REACHES database —-> This is an ambiguous expression. "mainly" should be replaced by "mostly" or "basically".

P.7 In Figure 1, the scale on the horizontal axis should be expressed as "1650 1660 1670 ——", not as "1644 1654 1664 ——". This is the same for other figures; Fig.2, Fig.3, Fig.4. Strangely, Figure 5 on P.14 has no time scales. As for the time scale, Figure 7, 8, 9, 10, and 12 are good, but Fig.11 should be corrected.

3.Methods

P.9 Line 1 - 8: In this paragraph, the term "drought" is defined as divided into "meteorological drought" and "hydrological drought", but the distinction between the two is arbitrary and lacks objectivity.ãĂĂAuthors should give some text examples of meteorological droughts and hydrological droughts in the "Compendium of Chinese Meteorological Records of the Last 3000 Years", by citing specific sentences.

P.1 Line 12 - 14:ãĂĂThe term "Paoshan" should be defined; What's the difference

between "Paoshan of Shanghai City" and "period of Paoshan" ?

Line 23: "On very data" » "On every data"

P.11 Line 31: (1832——) » (1833——)

4.Results

P.12 Line 15 - 17: In figure 4A, the authors mention that "If taken drought variable as a major concern, there is only one spike around 1720 in the earlier half of the 18th century and some increasing frequency around 1730-1750.", this expression is subjective and inaccurate, especially for the term 1730-1750.

Line 17 - 23: This paragraph includes serious problems concerning the comparison between the time-series of multiple variables for droughts and the Northern Hemisphere temperature anomalies, as there exists a large difference of spatial and temporal scales between them. If the authors would discuss the relationship between the drought frequencies in China and the Northern Hemisphere mean temperature anomalies, reasonable explanations for the peak of drought frequency and the NH mean temperature anomalies in terms of anomalous atmospheric circulation patterns which might cause surface drought conditions in China.

P.14 The description of colored lines drawn in Figure 5 is not specified, except for Famine, Crop Failure and Socioeconomic turmoil. Also, the scale of the year on the horizontal axis of the graph in Figure 5. is completely missing.

P.14 Line 14 - P.15 Line 6: The description in this paragraph is arbitrary and less objective. For example, the authors identified six severe drought periods, but no explanations for the specific selection criteria can be found. In case of the period 1720-1740, the drought frequency in the 1730s was apparently lower than in the 1750s (Fig.4A) . So, please mention clearly the specific selection criteria for 6 severe drought periods.

Figure 6 (P.16-18): The size of the legend on the left in figures is too small to recognize. These should be more expanded for the readable size.

P.20 In Fig.7 and Fig.9, population changes should be shown as a line graph, not as a dotted graph.

Line 19: The authors mention that "the population of Jiangsu showed a downward (Figure 8)", but no population graph can be found in Figure 8.

P.22 Line 19: expended » expanded

P.26 5.Discussions and conclusions Line 18: metrological » meteorological Line 21: dry » dry conditions

P.27 Line 22: while » which ? Line 23: between it » between them ? Line 35: drought though can be –» though drought can be – ?

P.28: Line 30: in the 1665-1991 » in the 1665-1911 ?

P.29: Line 11 - 13: In this paragraph, the authors pointed out that "Moreover, this illustrates the importance to separately deal with drought and flood events instead of integrating them into one single index as practiced in many previous studies". Probably, the authors did not read at least two important papers below; the former article analyzed the spatiotemporal variations of droughts and floods in China during the historical period based on statistical analysis, and the latter article reviewed historical climate records in China and reconstruction of past climates. The authors should discuss by citing and referring these valuable papers.

Wang, S.W., and Z.C.Zhao, 1981: Droughts and floods in China, 1470-1979. in "Climate and History" T.M.L.Wigley, M.J.Ingram and G.Farmer (eds.), Cambridge University Press, 271-288.

Zhang, Jiacheng and T.J.Crowley, 1989: Historical climate records in China and reconstruction of past climates. Journal of Climate, 833-849.

---

## Author Comment (AC1) · 10 Jan 2020

This is an important topic in the research on past climate change. The topic also fits the journal. It cannot be accepted for publication, because there are lots of problems must be solved seriously. First, the manuscript selected the drought and reconstruction and its impact on human society. So far, there are lots of studies in this direction. May I know the innovative points of this manuscript, in comparison with these existing findings? The authors did not make it clear.

Response: There are indeed a lot of studies focusing on historical droughts in China, Asia, Europe, North America and other continents. In the manuscript (abstract and introduction), we have mentioned that tree rings, PDSI, and documentary records are

important sources for reconstructing drought. The longest and most popular approach in China especially for reconstructing drought from documentary sources is the use of the dryness-wetness index which represents a sophisticated and mature formula to judge records and assign records ordinal scale values to form the data (see page 3-4 of the manuscript). The index system is admirable and well established, however it also presents some difficulties for readers to understand how the initial records are interpreted, the statistics of the raw data (i.e. amount of the records), and the robustness of the interpretation, for instance. Our purpose in this study is to take advantage of the database and to develop a methodology that can enhance the transparency of data processing. So, the innovation is that it advances the methodology by displaying raw data profiles of the drought frequency and categorizing them into various groups through careful interpretation of the drought vocabularies recorded in the historical documents (e.g. pure drought vocabularies as meteorological drought, vocabularies with dried water body as hydrological drought). As shown in the manuscript, this methodology can largely increase the transparency of the raw data and test the robustness of the drought series by comparing different groups of drought records. For example, figure 2 and 3 illustrate good consistency between drought and severe drought and between meteorological drought and hydrological drought. To our knowledge, this study is among the pioneer studies of the like that endeavors to reveal and test drought records in the historical climatology field.

Second, there is a big gap between their aim and their methods. In the manuscript, "our objective is to make every drought and associated variables as literally clear and operationally independent as possible." How the authors could evaluate the objective of "clear and operationally independent"? This object has not been discussed in the later sections. Did authors achieve the aim? Please clarify. Response: As we mentioned in the manuscript (page 2-3), interpretation and identification of drought in the documentary records are controversial because it's often not clear the drought related to meteorological, hydrological, or other socioeconomic processes. In the meantime, previous practices of dryness-wetness or drought index generally integrate all those processes into five- or seven-scaled grading. For example, occurrence of locust plague, dried waterbody, crop failure and famine are all treated as criteria in grade judgement of the index system (Page 4 Line 19-27). The approach, while meaningful, presents a methodological difficulty to distinguish among those different processes and is not possible to statistically examine their correlations because all variables have been put in the composites. Thus in this study, we purposively define and separate different drought categories, and other variables like crop failure, famine, locust and socioeconomic turmoil to maintain their independency which will allow us to conduct cross check and statistical examination over their relations in the later analysis as shown in the Kernel density spatial pattern and network analysis.

Third, the authors are very proud of REACHES. I also read the paper introducing REACHES as shown in the reference of the manuscript. In fact, the database is from Compendium of Chinese Meteorological Records of the Last 3,000 Years (Zhang, 2013). This book is the basis for whole research and makes REACHES scientific and trustable. The authors should not over-emphasize the importance and innovation of REACHES. Response: We are thankful for this constructive comment. REACHES presents a digital database and so far now everything of it comes from the Compendium (Zhang 2013). We have made this clear in the manuscript. We shall be careful not to give readers such an impression of overriding the original source, in the meanwhile, we shall be responsible to deliberate on data retrieval from the REACHES and data processing. We will make modification on this point.

Fourth, as mentioned by the authors, "To comprehensively compare and analyze drought and associated data series from the REACHES with other socioeconomic variables from independent data sources, several archival and index data were also collected for analysis." May I know the similar or different features in the records of these documents? The authors did not discuss enough to compare these sources Response: The main body of the research is to deliver reliable drought and other associated variables' (e.g. crop failure, famine, locust) temporal trends and spatial patterns. Other

archival and index data were used to cross check the drought analysis and to provide insightful information to explain the severe drought events. Those archival and index databases were previously established by different independent organizations.This is what we meant here for independent data sources. On page 8 Line 11-23, we have explained their sources such as grain price data (based on monthly grain price report), civil war data (from Chronology of China's Ancient Wars), and population data (from several different sources ranging from Registers of Quantities of Provincial Population and Grain Storage to The History of Population in China). One might suspect if these data series across grain price, wars, population and the Compendium (which constitutes REACHES) could come from similar groups of historical books. Our response is that the initial sources of those databases seem to have their separate specialities. There might be some coverage for a certain degree, but we are not sure the percentage and we believe this is not the work for the present study.

Fifth, there are different categories of drought recorded in the historical documents. Why there are different records in the documents? Please clearly explain the reasons to have these different records in historical documents. Then, the readers will know rationale of these categorizations and see the importance of this research. Response: As mentioned before, 'drought' in historical documents is not a rigorous and conceptually straightforward word. (Page 3, Line 13-22) Several studies thus purpose four categories of drought: meteorological drought, agricultural drought, hydrological drought, and socioeconomic drought amplified by negative effect of drought to influence everyday life and social stability. All these definitions are explained in the manuscript. When ancient people wrote those records, they generally would describe what they saw of the phenomena and the related environmental and socioeconomic effects. This is why there are so many different descriptions of droughts phenomena in the records. So, to avoid mis-interpretation, we adopted the approach to seriously deal with and interpret the contents of the records and further categorize them instead of just putting them as same group definition of drought. We shall be more carefully discussing this in the revision.

Sixth, I am not convinced by the Kernel method. It is common to have the missing data in the documents among different regions. If using the number of records, the results will be disturbed by the data availability condition. The results are thus not reliable at all. Response: Kernel density is an equation that calculates every grid value by considering the values of the geographically adjacent grids by assign the weights (Page 10 for detailed explanation). It is a very mature and common approach for spatial analysis. The algorithm is based on the original distribution of the raw data and not to transform the data. Its strength is to make the distribution pattern clearer by data smoothening, meaning the effect of outliers and noises can be lowered. In this case, blank (while) area mostly reflects missing value (=0) or very low value. All data analysis is subject to data availability. And compared to record frequency distribution as also shown on figure 6 (right panel), Kernel density maps clearly present rigid spatial pattern of the interested phenomena than pure frequency distribution maps.

Seventh, in Page 11, there are some linkages according to the one record, "the events would be decomposed and then displayed in a way that drought linked with rainfall, drought linked with frost, drought linked with rice price, rainfall linked with rice price, frost linked with rice price and so on to further calculate their pairwise coefficients." It is not persuasive to have such findings by only one record. In fact, the authors should revise the whole manuscript to review their findings. Please avoid similar problems. Response: This sentence as mentioned by the reviewer is a bad organized sentence (Page 11, Line 34-37). This single line of record does mention several different phenomena 'occurring at different times of the year'. We will rewrite the sentence and make sure all algorithm is correct in the revision.

Eighth, please check the language. There are some typos. Such as, it should be Guang Ling but not Quang Ling in Page 11. Response: Thanks for the correction. We will revise and check the language.

Ninth, the language is not clear and concise enough. There are many redundant sentences in the manuscript, such as Page 2 "Studying past drought and humidity has

been a long practiced subject in historical climatology and paleoclimatology". If the authors still want to keep these redundant sentences, why not add the references? Response: We will check again the language and remove the redundant and unnecessary sentences to improve its quality.

In terms of reference, the authors have many judgements without the proofs from their articles. For example, Page 3 "Yet, tree ring reconstruction usually suffers from growing seasonality of trees and blurred interpretation of isotopes." This is not your findings from the manuscript. There are many similar examples in the manuscript to show the authors are not careful enough to conduct the research and claim their findings. Response: This comment is constructive; we will add references at appropriate places and avoid statements of the kind.

Please also note the supplement to this comment:
https://www.clim-past-discuss.net/cp-2019-115/cp-2019-115-AC1-supplement.pdf

---

## Author Comment (AC2) · 10 Jan 2020

GENERAL COMMENTS This article discusses historical droughts and the role of human interventions in the Qing Dynasty (1644-1911) of China based on the REACHES database, which was created using the "Compendium of Meteorological Records of China in the Last 3000 Years". The main purpose of this article would be an analysis of longterm variations of droughts and their impacts on human society in China during 1644-1911. Although the methodology used might be somewhat innovative, the results were not so new and interesting as compared with a lot of previous similar papers analyzing the changes in climate and natural disasters in China during the historical period. Also, another problem of this article is that four authors of this article are the same as those of the main referenced paper by Wang,P.K. et al.(2018) which

introduced the construction of the REACHES database. For most of the readers of this article, the REACHES database might be unfamiliar and unrecognized. Therefore, authors should explain the REACHES database in detail at first using visual diagrams and charts, such as shown in Wang,P.K. et al.(2018). However, this article would be acceptable for publication after minor revisions. Response: We thank for the many constructive comments. We will surely add a description about the REACHES database in the paper to help readers familiarize with the database.

SPECIFIC COMMENTS I. Introduction P.2 Line 26: Studying past drought and humidity Âż Studying past drought and flood After this sentence, previous related papers should be referred to. Line 34: documented records Âż historical documents (e.g., ****,*****) P.3 Line 30-31: Three-, five-, ———- is the most commonly practiced method (previous articles should be referred here), so that ——— Response: Thanks for the suggestions. We will correct those in revision.

2.Data P.5 Line 22: The data source of this study mainly comes from REACHES database —-> This is an ambiguous expression. "mainly" should be replaced by "mostly" or "basically". P.7 In Figure 1, the scale on the horizontal axis should be expressed as "1650 1660 1670 —–", not as "1644 1654 1664 ——". This is the same for other figures; Fig.2, Fig.3, Fig.4. Strangely, Figure 5 on P.14 has no time scales. As for the time scale, Figure 7, 8, 9, 10, and 12 are good, but Fig.11 should be corrected. Response: Thanks for the suggestions. We will correct those in t revision.

3.Methods P.9 Line 1 - 8: In this paragraph, the term "drought" is defined as divided into "meteorological drought" and "hydrological drought", but the distinction between the two is arbitrary and lacks objectivity. Authors should give some text examples of meteorological droughts and hydrological droughts in the "Compendium of Chinese Meteorological Records of the Last 3000 Years", by citing specific sentences. Response: This is very useful suggestion. We will do this in revision. P.1 Line 12 - 14:The term "Paoshan" should be defined; What's the difference between "Paoshan of Shanghai City" and "period of Paoshan" ? Response: Thanks for the careful review. Paoshan

is a district in Shanghai City. The latter one you mentioned here should be corrected as the frequency for the period 'in' Paoshan district... Line 23: "On very data" Âż "On every data"
P.11 Line 31: (1832——) Âż (1833——)
 4.Results P.12 Line 15 - 17: In figure 4A, the authors mention that "If taken drought variable as a major concern, there is only one spike around 1720 in the earlier half of the 18th century and some increasing frequency around 1730-1750.", this expression is subjective and inaccurate, especially for the term 1730-1750. Response: We also agree that the description is a bit arbitrary and subjective, and will modify the description. Line 17 - 23: This paragraph includes serious problems concerning the comparison between the time series of multiple variables for droughts and the Northern Hemisphere temperature anomalies, as there exists a large difference of spatial and temporal scales between them. If the authors would discuss the relationship between the drought frequencies in China and the Northern Hemisphere mean temperature anomalies, reasonable explanations for the peak of drought frequency and the NH mean temperature anomalies in terms of anomalous atmospheric circulation patterns which might cause surface drought conditions in China. Response: We agree that the comparison of drought frequencies in China and NH mean temperature anomalies is scale inappropriate. The initial idea is to illustrate a general pattern of warmer temperature anomaly in the 18th century corresponding to less drought frequencies in our records. We will rethink about how to reorganize the paragraph and give reasonable explanations. P.14 The description of colored lines drawn in Figure 5 is not specified, except for Famine, Crop Failure and Socioeconomic turmoil. Also, the scale of the year on the horizontal axis of the graph in Figure 5. is completely missing. Response: Thanks. We will definitely deal with this critical issue!

P.14 Line 14 - P.15 Line 6: The description in this paragraph is arbitrary and less objective. For example, the authors identified six severe drought periods, but no explanations for the specific selection criteria can be found. In case of the period 1720-1740, the drought frequency in the 1730s was apparently lower than in the 1750s (Fig.4A) . So, please mention clearly the specific selection criteria for 6 severe drought periods. Response: Although there are already some explanations about the selection of the six severe drought periods, we agree that the criteria can be more clear and quantitative. We will make revision about the criteria from both quantitative frequency and narrative analysis.

Figure 6 (P.16-18): The size of the legend on the left in figures is too small to recognize. These should be more expanded for the readable size. Response: We will modify this in revision. P.20 In Fig.7 and Fig.9, population changes should be shown as a line graph, not as a dotted graph. Response: The initial purpose is to show the original data. We can make it a line graph in revision. Line 19: The authors mention that "the population of Jiangsu showed a downward (Figure 8)", but no population graph can be found in Figure 8. Response: This one should be referred to Figure 7.

P.22 Line 19: expended Âż expanded P.26 5.Discussions and conclusions Line 18: metrological Âż meteorological Line 21: dry Âż dry conditions P.27 Line 22: while Âż which ? Line 23: between it Âż between them ? Line 35: drought though can be –Âż though drought can be – ? P.28: Line 30: in the 1665-1991 Âż in the 1665-1911 ? Response: We thank for all above corrections!

P.29: Line 11 - 13: In this paragraph, the authors pointed out that "Moreover, this illustrates the importance to separately deal with drought and flood events instead of integrating them into one single index as practiced in many previous studies". Probably, the authors did not read at least two important papers below; the former article analyzed the spatiotemporal variations of droughts and floods in China during the historical period based on statistical analysis, and the latter article reviewed historical climate records in China and reconstruction of past climates. The authors should discuss by citing and referring these valuable papers. Wang, S.W., and Z.C.Zhao, 1981: Droughts and floods in China, 1470-1979. in "Climate and History" T.M.L.Wigley, M.J.Ingram and G.Farmer (eds.), Cambridge University Press, 271-288. Zhang, Jiacheng and T.J.Crowley, 1989: Historical climate records in China and reconstruction of past climates. Journal of Climate, 833-849. Response: Drought and

Flood Charts of the five hundred years and the two publications mentioned here are all influential works. While the methodologies are different, we agree that we could revise the sentence and more considerably discuss what messages we want to deliver here. Thank you.

Please also note the supplement to this comment:
https://www.clim-past-discuss.net/cp-2019-115/cp-2019-115-AC2-supplement.pdf
* * *

---

## Author Response (AR1)

**Historical droughts in the Qing dynasty (1644–1911) of China () (cp-2019-115)**

**Overall reply**

We thank for both referees' constructive comments. We have considerably revised the manuscript according to the comments. Some of the parts, especially results and discussions and conclusions sections, have been rewritten to improve their structure and readability. In addition to that, we have also suggested to mend the title, removing the point of 'the role of human intervention'. This is because we hope to focus the line on the historical drought and the methods that we developed for drought analysis. We hope to deliver more in-depth analysis on drought and societal interaction in the future study. Below are the point by point responses to the referees' comments.

**Response to Referee#1**
**Note: Referee comments are shown in black color and our responses in blue.**

This is an important topic in the research on past climate change. The topic also fits the journal. It cannot be accepted for publication, because there are lots of problems must be solved seriously. First, the manuscript selected the drought and reconstruction and its impact on human society. So far, there are lots of studies in this direction. May I know the innovative points of this manuscript, in comparison with these existing findings? The authors did not make it clear.

Response: There are indeed a lot of studies focusing on historical droughts in China, Asia, Europe, North America and other continents. In the manuscript (abstract and introduction), we have mentioned that tree rings, PDSI, and documentary records are important sources for reconstructing drought. The longest and most popular approach in China especially for reconstructing drought from documentary sources is the use of the dryness-wetness index which represents a sophisticated and mature formula to judge records and assign ordinal scale values to form the data (see page 3-4 of the manuscript). The index system is admirable and well established, however it also presents some difficulties for readers to understand how the initial records are interpreted, the statistics of the raw data (i.e. amount of the records), and the robustness of the interpretation, for instance.

Our purpose in this study is to take advantage of the database and to develop a methodology that can enhance the transparency of data processing. So, the innovation is that it advances the methodology by displaying raw data profiles of the drought frequency and categorizing them into various groups through careful interpretation of the drought vocabularies recorded in the historical documents (e.g. pure drought vocabularies as meteorological drought, vocabularies with dried water body as hydrological drought). As shown in the manuscript, this methodology can largely increase the transparency of the raw data and test the robustness of the drought series by comparing different groups of drought records. For example, figure 2 and 3 illustrate good consistency between drought and severe drought and between meteorological drought and hydrological drought. To our

knowledge, this study is among the pioneer studies of the like that endeavor to reveal and test drought records in the historical climatology field.

Second, there is a big gap between their aim and their methods. In the manuscript, "our objective is to make every drought and associated variables as literally clear and operationally independent as possible." How the authors could evaluate the objective of "clear and operationally independent"? This object has not been discussed in the later sections. Did authors achieve the aim? Please clarify.

Response: As we mentioned in the manuscript (page 2-3), interpretation and identification of drought in the documentary records are controversial because it's often not clear whether the drought related to meteorological, hydrological, or other socioeconomic processes. In the meantime, previous practices of dryness-wetness or drought index generally integrate all those processes into five- or seven-scaled grading. For example, occurrence of locust plague, dried waterbody, crop failure and famine are all treated as criteria in grade judgement of the index system (section 2). The approach, while meaningful, presents a methodological difficulty to distinguish among those different processes and is not possible to statistically examine their correlations because all variables have been put in the composites. Thus in this study, we purposively define and separate different drought categories, and other variables like crop failure, famine, locust and socioeconomic turmoil to maintain their independency which allows us to conduct cross check and statistical examination over their relations in the later analysis as shown in the Kernel density spatial pattern and network analysis. And importantly, we have also clearly reveal our methods of data retrieval and statistics in the paper (section 3.1).

Third, the authors are very proud of REACHES. I also read the paper introducing REACHES as shown in the reference of the manuscript. In fact, the database is from Compendium of Chinese Meteorological Records of the Last 3,000 Years (Zhang, 2013). This book is the basis for whole research and makes REACHES scientific and trustable. The authors should not over-emphasize the importance and innovation of REACHES.

Response: We are thankful for this constructive comment. REACHES presents a digital database and so far now everything of it comes from the Compendium (Zhang 2013). We have added more sentences in the beginning of the data section 3.1 to highlight this. We shall be careful not to give readers such an impression of overriding the original source, in the meanwhile, we shall be responsible to deliberate on data retrieval from the REACHES and data processing.

Fourth, as mentioned by the authors, "To comprehensively compare and analyze drought and associated data series from the REACHES with other socioeconomic variables from independent data sources, several archival and index data were also collected for analysis." May I know the similar or different features in the records of these documents? The authors did not discuss enough to compare these sources

Response: The main body of the research is to deliver reliable drought and other associated variables' (e.g. crop failure, famine, locust) temporal trends and spatial patterns. Other archival and index data were used to cross check the drought analysis and to provide insightful information to

explain the severe drought events. Those archival and index databases were previously established by different independent organizations. This is what we meant here for independent data sources. In section 3.2, we have explained their sources. One might suspect if these data series across grain price, wars, population and the *Compendium* could come from similar groups of historical books. Our response is that the initial sources of those databases seem to have their separate specialties. There might be some overlap for a certain degree, but we are not sure the percentage and we believe this is not the work for the present study. However, we add some lines to stress the point "Practically, it might be unavoidable that some of the contents in the historical books, if referred to climatic and weather conditions, were quoted in the *Compendium*. For example, there are 5 quotations of records from Actual Veritable Records of Emperors of the Qing Dynasty and 148 records from Draft History of the Qing Dynasty found in the REACHES among the overall of 93,415 records in the Qing dynasty. The coverage, however, is minimal in quantity among all". Thank you for the constructive point.

Fifth, there are different categories of drought recorded in the historical documents. Why there are different records in the documents? Please clearly explain the reasons to have these different records in historical documents. Then, the readers will know rationale of these categorizations and see the importance of this research.

Response: As mentioned before, 'drought' in historical documents is not a rigorous and conceptually straightforward word. Several studies thus propose four categories of drought (Page 3, 2$^{nd}$ paragraph): meteorological drought, agricultural drought, hydrological drought, and socioeconomic drought amplified by negative effect of drought to influence everyday life and social stability. All these definitions are explained in the manuscript. When ancient people wrote those records, they generally would describe the phenomena they saw and the associated environmental and socioeconomic effects. This is why there are many different descriptions of the drought phenomena in the records. So, to avoid misinterpretation, we adopted the approach to seriously deal with and interpret the contents of the records and further categorize them instead of just putting them as same group definition of drought (see section 3.1).

Sixth, I am not convinced by the Kernel method. It is common to have the missing data in the documents among different regions. If using the number of records, the results will be disturbed by the data availability condition. The results are thus not reliable at all.

Response: Kernel density is an equation that calculates every grid value by considering the values of the geographically adjacent grids by assign the weights (Page 10 section 4.2 for detailed explanation). It is a very mature and common approach for spatial analysis. The algorithm is based on the original distribution of the raw data and not to transform the data. Its strength is to make the distribution pattern clearer by data smoothening, meaning the effect of outliers and noises can be lowered. In this case, blank (while) area mostly reflects missing value (=0) or very low value. All data analysis is subject to data availability. And compared to record frequency distribution (such as Figure 7 (b) social turmoil variable), Kernel density maps clearly present rigid spatial pattern of the interested phenomena than pure frequency distribution maps.

Seventh, in Page 11, there are some linkages according to the one record, "the events would be decomposed and then displayed in a way that drought linked with rainfall, drought linked with frost, drought linked with rice price, rainfall linked with rice price, frost linked with rice price and so on to further calculate their pairwise coefficients." It is not persuasive to have such findings by only one record. In fact, the authors should revise the whole manuscript to review their findings. Please avoid similar problems.

Response: This sentence as mentioned by the reviewer was a poorly organized sentence (Page 11, Line 34-37). We have rewritten this sentence. Notably, the original quotation does mention several different events occurred in the year, and we have carefully checked the reasonability of the method and the description.

Eighth, please check the language. There are some typos. Such as, it should be Guang Ling but not Quang Ling in Page 11.

Response: Thanks for the correction. We have modified this specific county name and revised/rewritten the manuscript.

Ninth, the language is not clear and concise enough. There are many redundant sentences in the manuscript, such as Page 2 "Studying past drought and humidity has been a long practiced subject in historical climatology and paleoclimatology". If the authors still want to keep these redundant sentences, why not add the references?

Response: We have thoroughly revised the manuscript to avoid the redundant sentences. For this specific one, we still suggest to keep it as a beginning sentence for the paragraph that focuses on historical climatology. Thank you for this insightful comment.

In terms of reference, the authors have many judgements without the proofs from their articles. For example, Page 3 "Yet, tree ring reconstruction usually suffers from growing seasonality of trees and blurred interpretation of isotopes." This is not your findings from the manuscript. There are many similar examples in the manuscript to show the authors are not careful enough to conduct the research and claim their findings.

Response: We thank for this comment. More references are added for the appropriateness to avoid the kind of ambiguities.

**Response to Referee#2**

GENERAL COMMENTS This article discusses historical droughts and the role of human interventions in the Qing Dynasty (1644-1911) of China based on the REACHES database, which was created using the "Compendium of Meteorological Records of China in the Last 3000 Years". The main purpose of this article would be an analysis of long term variations of droughts and their impacts on human society in China during 1644-1911. Although the methodology used might be

somewhat innovative, the results were not so new and interesting as compared with a lot of previous similar papers analyzing the changes in climate and natural disasters in China during the historical period.

Also, another problem of this article is that four authors of this article are the same as those of the main referenced paper by Wang,P.K. et al.(2018) which introduced the construction of the REACHES database. For most of the readers of this article, the REACHES database might be unfamiliar and unrecognized. Therefore, authors should explain the REACHES database in detail at first using visual diagrams and charts, such as shown in Wang,P.K. et al.(2018). However, this article would be acceptable for publication after minor revisions.

Response: We thank for the constructive comments. In the section 3.1 we have mended to add more sentences to introduce the REACHES in the meantime making it clear that the quotations are from the *Compendium*, as suggested by reviewer #1. However, we decided not to use diagrams and charts in the manuscript for introducing REACHES, since there are already 12 figures in the present manuscript. Instead, we add a sentence to emphasize that 'readers are referred to Wang et al., 2018 for detailed descriptions about the database (page 5).'

SPECIFIC COMMENTS

I. Introduction

P.2 Line 26: Studying past drought and humidity » Studying past drought and flood After this sentence, previous related papers should be referred to. Line 34: documented records » historical documents (e.g., ****,*****)

Response: Thanks. the reference has been added for all appropriateness.

P.3 Line 30-31: Three-, five-, ———- is the most commonly practiced method (previous articles should be referred here), so that ———

Response: Thanks for the suggestions. We have added the references.

2.Data

P.5 Line 22: The data source of this study mainly comes from REACHES database —-> This is an ambiguous expression. "mainly" should be replaced by "mostly" or "basically".

Response: Thanks. We have corrected it as 'Finally, and importantly, all records of the REACHES are taken from the Compendium of Chinese Meteorological Records of the Last 3,000 Years (Zhang, 2013)(page 6 last paragraph in revision)'

P.7 In Figure 1, the scale on the horizontal axis should be expressed as "1650 1660 1670 ——", not as "1644 1654 1664 ——". This is the same for other figures; Fig.2, Fig.3, Fig.4. Strangely, Figure 5 on P.14 has no time scales. As for the time scale, Figure 7, 8, 9, 10, and 12 are good, but Fig.11 should be corrected.

Response: The starting year of the Qing dynasty is 1644. That is why the time frame of the figures is generally from that year and shown in decadal scale. We hope this is acceptable. Thanks. All figures are checked. We hope the formatting problem while submission will not appear.

3.Methods

P.9 Line 1 - 8: In this paragraph, the term "drought" is defined as divided into "meteorological drought" and "hydrological drought", but the distinction between the two is arbitrary and lacks objectivity. Authors should give some text examples of meteorological droughts and hydrological droughts in the "Compendium of Chinese Meteorological Records of the Last 3000 Years", by citing specific sentences.

Response: This is very useful suggestion. Examples for meteorological and hydrological droughts are carefully explained in the section 4.1 (page 9).

P.1 Line 12 - 14:The term "Paoshan" should be defined; What's the difference between "Paoshan of Shanghai City" and "period of Paoshan" ?

Response: Thanks for the careful review. Paoshan is a district in Shanghai City (page 10 last paragraph). The latter one you mentioned should be corrected as the frequency for the period of Paoshan district. Nonetheless, we have modified Paoshan to Baoshan for the exact Chinese pinyin.

Line 23: "On very data" » "On every data"
P.11 Line 31: (1832——) » (1833——)

Response: Thanks. This was corrected (page 12 first paragraph).

4.Results

P.12 Line 15 - 17: In figure 4A, the authors mention that "If taken drought variable as a major concern, there is only one spike around 1720 in the earlier half of the 18th century and some increasing frequency around 1730-1750.", this expression is subjective and inaccurate, especially for the term 1730-1750.

Response:  we agree that many of these descriptions were debatable and unprecise. Thus, we have rewritten the whole results part to avoid the subjective explanations.

Line 17 - 23: This paragraph includes serious problems concerning the comparison between the time series of multiple variables for droughts and the Northern Hemisphere temperature anomalies, as there exists a large difference of spatial and temporal scales between them. If the authors would discuss the relationship between the drought frequencies in China and the Northern Hemisphere mean temperature anomalies, reasonable explanations for the peak of drought frequency and the NH mean temperature anomalies in terms of anomalous atmospheric circulation patterns which might cause surface drought conditions in China.

Response: We agree that the comparison of drought frequencies in China and NH mean temperature anomalies is scale inappropriate. The initial idea is to illustrate a general pattern of warmer temperature anomaly in the 18th century corresponding to less drought frequencies in our records. This is a general description, not a major point of the paper.

P.14 The description of colored lines drawn in Figure 5 is not specified, except for Famine, Crop Failure and Socioeconomic turmoil. Also, the scale of the year on the horizontal axis of the graph in Figure 5. is completely missing.

Response: Thanks for the note. There were apparently some errors when submission. We hope this is corrected in the revision.

P.14 Line 14 - P.15 Line 6: The description in this paragraph is arbitrary and less objective. For example, the authors identified six severe drought periods, but no explanations for the specific selection criteria can be found. In case of the period 1720-1740, the drought frequency in the 1730s was apparently lower than in the 1750s (Fig.4A) . So, please mention clearly the specific selection criteria for 6 severe drought periods.

Response: There were already some explanations about the selection of the six drought periods. However, in order to further justify its reasonability, we added a statistical description by screening the top 3% drought years. Please see revision page 10 first paragraph and pages 15-16.

Figure 6 (P.16-18): The size of the legend on the left in figures is too small to recognize. These should be more expanded for the readable size.

Response: Thanks. This has been remedied.

P.20 In Fig.7 and Fig.9, population changes should be shown as a line graph, not as a dotted graph.

Response: The initial purpose is to show the original data. But since the incompleteness of the data, we have decided to remove those.

Line 19: The authors mention that "the population of Jiangsu showed a downward (Figure 8)", but no population graph can be found in Figure 8.

Response: This one should be referred to Figure 7 in previous version. But since the incompleteness of the data, we have decided to remove those.

P.22 Line 19: expended » expanded

P.26 5.Discussions and conclusions Line 18: metrological » meteorological Line 21: dry » dry conditions

P.27 Line 22: while » which ? Line 23: between it » between them ? Line 35: drought though can be –» though drought can be – ?

P.28: Line 30: in the 1665-1991 » in the 1665-1911 ?

Response: We thank for all above corrections!

P.29: Line 11 - 13: In this paragraph, the authors pointed out that "Moreover, this illustrates the importance to separately deal with drought and flood events instead of integrating them into one single index as practiced in many previous studies". Probably, the authors did not read at least two important papers below; the former article analyzed the spatiotemporal variations of droughts and floods in China during the historical period based on statistical analysis, and the latter article reviewed historical climate records in China and reconstruction of past climates. The authors should discuss by citing and referring these valuable papers.

Wang, S.W., and Z.C.Zhao, 1981: Droughts and floods in China, 1470-1979. in "Climate and History" T.M.L.Wigley, M.J.Ingram and G.Farmer (eds.), Cambridge University Press, 271-288.

Zhang, Jiacheng and T.J.Crowley, 1989: Historical climate records in China and reconstruction of past climates. Journal of Climate, 833-849.

Response: Drought and Flood Charts of the five hundred years  (CMA 1981) and the two publications mentioned here are all influential works. We have previously reviewed those papers, and have cited them in appropriate places in the revision. More specifically, we have added the sentences to emphasize the contributions of them on page 4 first paragraph of section 2 and page 20 last paragraph. Thank you very much for the suggestions!

---

## Author Response (AR2)

**Historical droughts in the Qing dynasty (1644–1911) of China**

**Overall reply**

We sincerely thank for editor's and reviewers' positive comments and careful review of the manuscript. Below are the point by point response. Please note that all modification in the revision is highlighted in blue color.

**Comments and responses**
**Note: Review comments are shown in black color and our responses in blue.**

1) Please read carefully and follow instruction of Climate of the Past for the formal preparation of your manuscript – many points and parts of your manuscript does not follow them. The manuscript has to be carefully formally revised.
Response: Thanks for this. We have carefully remedied the format of the manuscript following the manuscript preparation guidelines.

2) Since authors claim that the meteorological drought, agricultural drought, hydrological drought and socioeconomic drought are the key innovative findings, I did not see the series of each category in the manuscript. The authors should list them and compare them with your adopted indicators to dig out the linkage in between. How four kinds of drought fluctuated in historical China? How four kinds of drought are connected with other disasters as mentioned?
Response: We thank for this valuable comment. In our manuscript, the operational definition of the four types of drought have been precisely defined in the Sect. 2.1 and 3.1, where we address that 'dried waterbody (representing hydrological drought), crop failure (representing agricultural drought), famine and socioeconomic turmoil (along with famine to represent socioeconomic drought).' (lines 134-136). The fluctuations of the series are shown in Figure 1, 2 &4, and the correlation coefficient can be found in Table 2. All the text related to this comment is highlighted in yellow color for editor and referee to check. We hope this can substantially increase the clarity.

Importantly, our purpose in this paper is to stress that careful deciphering of the records is very helpful to identify different processes related to drought phenomena, and the method is imperative for cross checking and identifying the correlations among them. Meanwhile, we do not want to over-emphasize the correlation coefficients discovered in this study since, as we mentioned in the lines 171-176 'some socioeconomic events that are not explicitly linked to climate conditions may not be included in the *Compendium*…This means that care must be taken when using and analyzing the socioeconomic variables in the REACHES because of the potentially biased sampling.' Therefore, there is a need in future study to use independent data series to further elaborate on the research subject.

3) It is suggested to the authors to share the data series as an attachment for free download. The attachment should be a computer-readable file. It will also increase the paper publicity.
Response: Yes, our idea is also to publicize the data series. We can provide them as attachment or supplementary to the manuscript (seemingly not suggested according to the guidelines), and deposit it at a FAIR-aligned data repository.

4) In Introduction you mention potential of reconstruction of drought indices (PDSI) from tree-ring data (Old World Drought Atlas). It would be correct to say somewhere, that similar high-quality quantitative drought-indices series (SPI, SPEI, Z index, PDSI) can be reconstructed from documentary data as shown, for example for the Czech Lands (Brázdil et al. 2016, Možný et al.

2016) with monthly, seasonal, half-year or annual resolution. Moreover, some other special drought indices series were developed in Europe like Drought Rogation Index in Spain (Barriendos 1997) or Drought Index in Italy (Diodato and Belocchi 2011). Brázdil et al. 2016 https://doi.org/10.3354/cr01380 Možný et al. 2016 https://doi.org/10.3354/cr01423 Barriendos 1997 https://doi.org/10.1177/ 095968369700700110 Diodato and Belocchi 2011 https://doi.org/10.3354/cr01020

Response: We highly appreciate the very informative comments from the editor, and have added those in the text as shown in lines 59-65. References related to this point are also added in the reference section.

5) Lines 118-171: Please include Section 2 as logic continuation of Introduction – you characterised situation with drought generally and now you are giving the Chinese results. Then Section 2 Data should follow etc. Lines 128-143: I recommend you to include this description of Grades 1-5 into simple table.

Response: We thank for the constructive suggestion and have made the corresponding modifications.

6) Section 6: Please try to separate this section to two separate sections Discussion and Conclusions. Conclusions in a few several point should shortly summarise your main results, which should be clearly communicated in the context of existing Chinese papers (what is new) and also as a contribution to drought studies in historical climatology in the international scale (e.g. with respect to the context of Brázdil et al. 2018 paper and recommended future research directions mentioned there).

Response: We thank for this suggestion. We have separated them into section 6 and section 7.

7) Please add also other requested points into manuscript (Data availability, Author contributions etc.) – see instructions for authors.

Response: This has been added in the revision. Thanks.

8) Authors in quotations in the text should be listed according to the year of publication, not according to their alphabet.

Response: Thanks. This has been also carefully corrected.

9) Please check your References if everything is quoted in the requested style – there are several quoted differently.

Response: Thank you. All references have been carefully checked for the format.

10) Examples of needed formal corrections:
Lines 75, 89 etc. - please use instead of documented "documentary" Line 144: Why not only "REACHES database" Figure 1: if you have in caption 1a and 1b, please put (a) and (b) directly into figures and delete "upper panel" and "lower panel" from figure caption. Figs. 2, 3 and further: Please use in description of y-axes "Number of …" not only N Line 385: please add to http when it was accessed Line 411: famers or farmers? Line 454: 2019p? Line 460: Frank et al. (2010). Lines 563-564 and further figures: is it correct … drought 7(a) and … 7(b) … 7(c)?

Response: Thank you so much for correcting the errors. These have all been modified and careful checks have also been performed throughout the manuscript.